*Report*

**EMBO** *reports*

# SETD2 methyltransferase activity promotes correct transcription initiation and termination

Magda Kopczyńska [1,2,8], Chihiro Nakayama [3,4,8], Agata Stępień [1,2], Shoko Ito [3], Koshi Imami [5], Michał R Gdula [1,6], Takayuki Nojima [3,7 ✉] & Kinga Kamieniarz-Gdula [1,2 ✉]

## Abstract

SETD2 is a methyltransferase responsible for depositing histone H3 lysine 36 trimethylation (H3K36me3). Loss of its enzymatic activity occurs in some cancers, including renal cell carcinoma. SETD2 mutations have been linked to delayed transcription termination but have not been explored in depth. Here, using nascent transcriptomics in SETD2 knockout and patient-derived cells, we reveal a dichotomy in SETD2 functions depending on the affected protein-coding gene. The majority of genes, named class I, are dependent on SETD2 function for transcription initiation, yet terminate transcription in the usual locations. In contrast, for class II genes, corresponding to 15–25% of active protein-coding genes, transcription initiation is robust in absence of SETD2 activity; however, widespread transcriptional readthrough occurs. Defective termination following SETD2 loss/mutation is associated with increased cryptic transcription initiation and impaired 3′ pre-mRNA cleavage. Additionally, alternative polyadenylation upon SETD2 activity loss is highly cell type specific, and no relationship with transcription readthrough was observed. We demonstrate that methyltransferase activity of SETD2 stimulates proper initiation, prevents cryptic initiation and promotes efficient 3′ end processing, however, it does so indirectly.

**Keywords** SETD2; Transcription Termination; Readthrough; Pre-mRNA 3' Cleavage; Cryptic Initiation
**Subject Category** Chromatin, Transcription & Genomics

## Introduction

Nuclear processes in eukaryotes like DNA replication and RNA transcription are tightly linked to chromatin structure and its modifications (Wang and Helin, 2025). Some epigenetic marks are stable and likely to be copied onto both sister chromatids to preserve gene expression patterns in between parental cells and daughter cells (Fitz-James and Cavalli, 2022; Krabbe, 2024). Other marks, such as trimethylation of histone H3 lysine 4 (H3K4me3) are relatively fragile and involved in quality control of RNA polymerase II (RNAPII) transcription at the promoter proximal region (Sun et al, 2023; Wang et al, 2023). Among the dynamic chromatin marks trimethylation of lysine 36 on histone H3 (H3K36me3) stands out due to its distribution along actively transcribed gene bodies. H3K36me3 is deposited co-transcriptionally by SETD2 that binds to SPT6 and the C-terminal domain of actively transcribing RNAPII (Li et al, 2005; Yoh et al, 2008; Markert et al, 2025). H3K36me3 profile covers the entire gene body, gradually increasing from 5' to 3' end, peaking at the polyadenylation site (PAS) (Edmunds et al, 2008). Therefore, SETD2 establishes a memory of active RNAPII transcription in chromatin via histone mark H3K36me3 (Kizer et al, 2005; Markert et al, 2025). So far, SETD2 has been reported to be a sole histone methyltransferase in mammals responsible for H3K36me3 deposition (Sun et al, 2005). Besides its role in marking active transcription units, is has been reported that via H3K36me3 deposition SETD2 regulates alternative splicing (Luco et al, 2010; Iwamori et al, 2016; Pradeepa et al, 2012; Bhattacharya et al, 2021; Zhu et al, 2017), DNA damage repair (Pfister et al, 2014; Miller et al, 2023) and chromatin organization (Wagner and Carpenter, 2012; Xie et al, 2022, 2023). Notably, SETD2 is among the frequently mutated epigenetic factors in a variety of human cancers. Around 15% of renal cell carcinoma (RCC) patients bear SETD2 catalytic mutations (Peña-Llopis et al, 2013).

Transcription by RNAPII is a multi-stage process and each step —initiation, pausing, elongation, and termination—is influenced by the chromatin context. RNAPII is recruited to gene promoters and initiates RNA synthesis; next, it pauses 20–40 nucleotides downstream of transcription start site (TSS) (Fong et al, 2022) and is released to productive RNA elongation (Core and Adelman, 2019). After RNAPII passes the polyadenylation signal at the 3' end of the gene, the newly synthesized pre-mRNA is cleaved by the cleavage and polyadenylation complex (Barillà et al, 2001; Takagaki and

[1]Center for Advanced Technologies, Adam Mickiewicz University, Uniwersytetu Poznanskiego 10, 61-614 Poznan, Poland. [2]Department of Molecular and Cellular Biology, Institute of Molecular Biology and Biotechnology, Faculty of Biology, Adam Mickiewicz University, Uniwersytetu Poznanskiego 6, 61-614 Poznan, Poland. [3]Medical Institute of Bioregulation, Kyushu University, 3-1-1 Maidashi, Higashi-ku, Fukuoka 812-8582, Japan. [4]Department of Medical Science, Kyushu University, 3-1-1 Maidashi, Higashi-ku, Fukuoka 812-8582, Japan. [5]Proteome Homeostasis Research Unit, RIKEN Center for Integrative Medical Sciences (IMS), Yokohama, Kanagawa, Japan. [6]Department of Gene Expression, Institute of Molecular Biology and Biotechnology, Faculty of Biology, Adam Mickiewicz University, Uniwersytetu Poznanskiego 6, 61-614 Poznan, Poland. [7]Present address: School of Medicine, Yokohama City University, 236-00043-9 Fukuura, Kanazawa-ku, Yokohama, Kanagawa, Japan. [8]These authors contributed equally: Magda Kopczyńska, Chihiro Nakayama. ✉E-mail: nojima.takayuki.058@m.kyushu-u.ac.jp; kinga.kamieniarz-gdula@amu.edu.pl

Manley, 1997; Meinhart and Cramer, 2004). RNAPII continues to transcribe DNA into RNA for several hundred to thousands of bases downstream of the PAS, and as it dissociates from the DNA template, transcription is terminated. Knockdown or inhibition of the endonuclease that cleaves nascent RNA, CPSF73, disrupts termination and induces transcriptional readthrough (Davidson et al, 2024; Eaton et al, 2020). However, termination can also occur prematurely within the gene body, regulated by factors such as the CPA component PCF11 (Kamieniarz-Gdula et al, 2019; Wang et al, 2019), and the Integrator complex (Skaar et al, 2015; Tatomer et al, 2019; Elrod et al, 2019; Huang et al, 2020) in protein-coding genes, as well as the Restrictor complex in non-coding transcripts (Estell et al, 2021; Rouvière et al, 2023; Nojima and Proudfoot, 2022; Austenaa et al, 2015). Moreover, most eukaryotic genes harbor multiple PASs, leading to alternative polyadenylation (APA) that results in expression of multiple gene isoforms (Mitschka and Mayr, 2022). Although APA determines mRNA 3' end, the link between APA usage and transcription termination remains unexplored.

Previous studies in budding yeast showed that loss of Set2 (SETD2 homolog) or H3K36me3 mark led to upregulation of transcripts derived from cryptic transcription initiation sites. This indicates that H3K36me3 prevents spurious initiation (Kaplan et al, 2003; Li et al, 2007). In mammals, H3K36me3 recruits DNA methyltransferases to the gene body, and gene body DNA methylation in turn prevents aberrant transcription initiation (Baubec et al, 2015; Neri et al, 2017). H3K36me3 is therefore a conserved platform for recruiting downstream factors that maintain proper transcription initiation across eukaryotes. Recent studies demonstrate that alternative TSS usage is connected to PAS usage, suggesting a functional coupling between gene 5'end and 3' end (Alfonso-Gonzalez and Hilgers, 2024; Alfonso-Gonzalez et al, 2023; preprint: Calvo-Roitberg et al, 2024). Importantly, total RNA-seq analyses of RCC patient tumor bearing SETD2 catalytic mutations detected chimeric transcripts across tandem transcription units. This result indicates that H3K36me3 prevents readthrough transcription and intergenic splicing (Grosso et al, 2015). Conventional RNA-seq was also used to report that SETD2 downregulation leads to cryptic initiation within gene bodies; however, this method captures only mature RNA, and the evidence was therefore indirect (Carvalho et al, 2013). Taken together, these findings suggest that the loss of SETD2 and/or H3K36me3 perturbs the transcription cycle. However, none of this has been supported by nascent RNA sequencing approaches. Due to technical limitations and reliance on indirect measurements in previous studies, the mechanisms by which SETD2 and/or H3K36me3 maintain productive RNAPII transcription in human cells remain largely unclear.

To dissect SETD2's role, we leveraged nascent RNA-sequencing technologies. mNET-seq provides single-nucleotide resolution of RNAPII activity (Nojima et al, 2015), that can be combined with a specific state of C-terminal domain (CTD) phosphorylation of RNAPII (Schlackow et al, 2017). In particular, mapping RNAPII phosphorylated at threonine 4 of its CTD marks specifically transcription termination sites (Kopczyńska et al, 2025). POINT-seq profiles whole nascent RNA, thus precisely maps transcription start sites and co-transcriptional RNA processing (Sousa-Luís et al, 2021). These tools allow us to pinpoint SETD2-dependent changes across the transcription cycle.

In this study, we demonstrate that SETD2 knockout or catalytic mutation disrupts both initiation and termination of RNAPII transcription in human cells, revealing two distinct classes of genes with SETD2 dependency. We uncover cryptic initiation and transcriptional readthrough in SETD2-deficient cells, as well as suppression of RNA cleavage at polyadenylation sites. Importantly, our findings reveal that SETD2 activity safeguards transcription termination indirectly, highlighting its role in mediating epigenetic regulation with 3' end processing. By integrating nascent transcriptomics with a comprehensive analysis, we provide compelling evidence that SETD2 maintains transcriptional fidelity, offering new perspectives on how chromatin modifications orchestrate distinct transcription stages.

## Results and discussion

### SETD2 contributes to proper gene definition by stimulating transcription initiation at the 5' end and preventing transcription readthrough at the 3' end

It has been previously shown that SETD2 loss can lead to RNAPII transcription running further downstream of genes (transcription readthrough) (Grosso et al, 2015). To better characterize and understand the effect of SETD2 on transcription, we took advantage of a powerful nascent transcriptomics method called polymerase-associated intact nascent transcripts (POINT) technology (Sousa-Luís et al, 2021). We first performed POINT-seq on U2OS cells with SETD2 knocked-out by CRISPR/Cas9 (KO) (Pfister et al, 2014), as well as on their wild-type (WT) counterparts (Fig. EV1A,B). Nascent RNA profiles revealed that most genes (which we named class I genes) showed no transcription readthrough upon SETD2 KO, and only a smaller subset of genes had extended transcription downstream of the PAS (Figs. 1A,B and EV1A). To systematically identify and analyze those genes, named class II, we defined them as genes where nascent RNA (POINT-seq) signal reached at least 1 kb further downstream in the SETD2 KO cells in comparison to the WT condition (Fig. EV1C). This analysis revealed that 24% of genes in U2OS cells showed transcription readthrough in SETD2 KO, and 76% did not (Fig. 1C,D). Interestingly, while class II genes showed a similar level of transcription initiation in WT and SETD2 KO conditions, indicated by comparable nascent RNA signal at TSS (Fig. 1D), class I genes showed on average decrease of transcription initiation (approximately by a third, Fig. 1C).

To corroborate that the transcription readthrough we observed is associated with perturbed transcription termination, we then measured RNAPII terminal pausing with the use of T4ph-mNET-seq method (Kopczyńska et al, 2025) (Fig. EV1D). For class I genes, the T4ph-mNET-seq signal was reduced (Fig. 1E), consistent with a decrease in nascent transcription observed on those genes (Fig. 1C). In contrast, for class II genes, the RNAPII terminal pausing window extended further downstream of the PAS in SETD2 KO versus WT condition (Fig. 1F), confirming the results of POINT-seq. T4ph-mNET-seq analysis also revealed that the transcriptional readthrough upon SETD2 loss is associated with RNAPII entering the terminal pausing mode at the same positions as in the WT situation (initial overlap in T4ph mNET-seq signal downstream of PAS for class II genes, Fig. 1F). This is different to readthrough that occurs

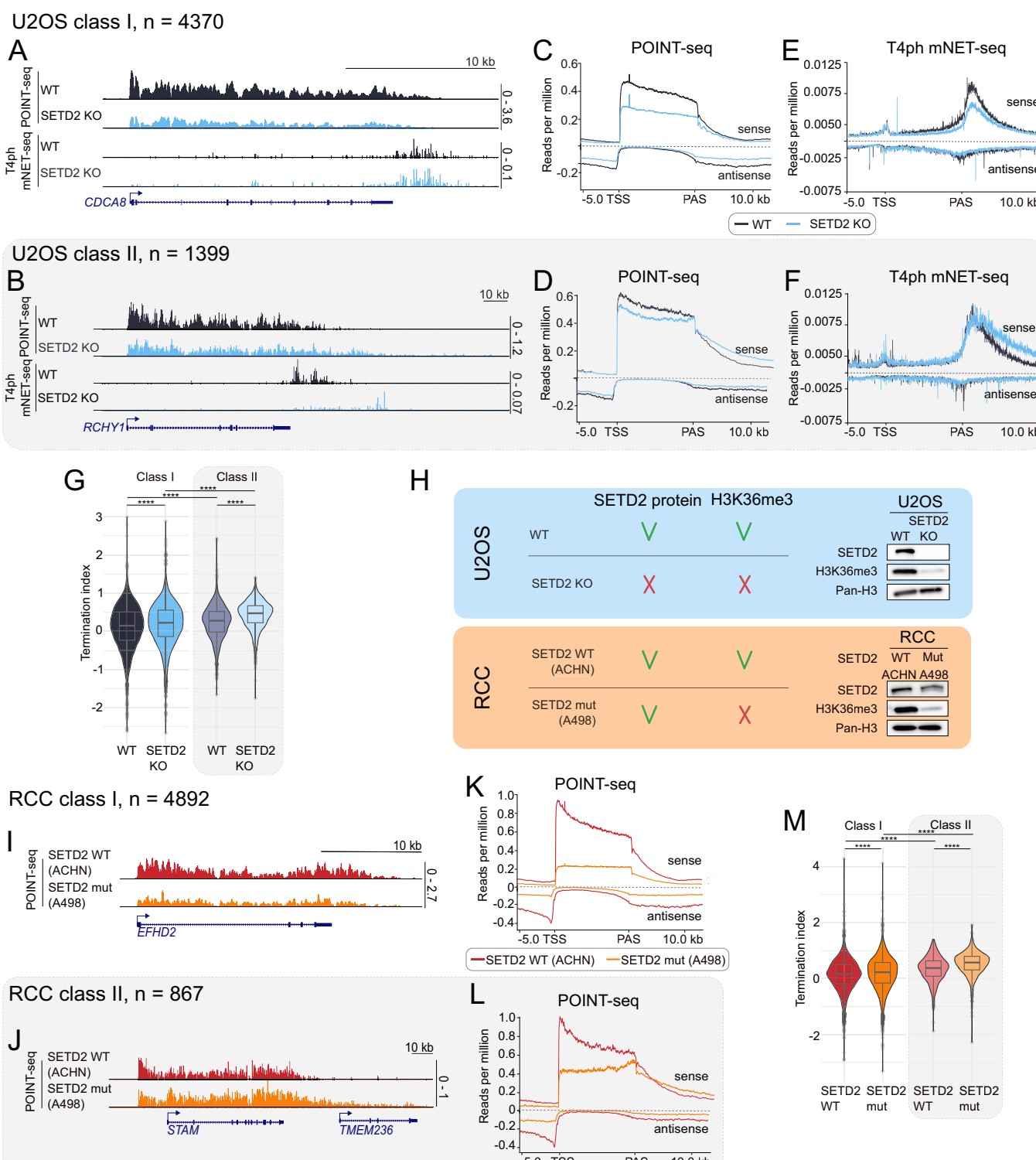

as a result of depletion of pre-mRNA 3' cleavage or termination factors, which instead leads to delayed RNAPII pausing and a shift of the entire RNAPII termination window downstream of the PAS (Schlackow et al, 2017; Kamieniarz-Gdula et al, 2019).

While class I genes did not show readthrough transcription, the nascent RNA signal profile (Fig. 1A,C) was suggestive of defective

transcription termination also on those genes, masked by decreased global transcriptional activity. To check and quantify a potential increase of nascent transcription in termination regions relative to their upstream genic regions, we calculated the termination index (Fig. 1G). This confirmed that the termination index increased in SETD2 KO conditions relative to WT for both class I (Fig. 1G,

**Figure 1. Methyltransferase activity of SETD2 stimulates transcription initiation and prevents transcription readthrough.**

(A, B) POINT-seq and T4ph mNET-seq signal on gene examples from class I (A) and class II (B), U2OS WT (black) and SETD2 KO (blue). (C, D) POINT-seq metagene profiles for class I (C) and class II (D) genes. (E, F) T4ph mNET-seq metagene profiles for class I (E) and II (F) genes. (G) Termination index calculated for class I (left panel) and class II (right panel, brighter colors and gray shading) genes in U2OS WT and SETD2 KO cells. (H) Comparison of the abundance of SETD2 protein and H3K36me3 histone mark between the two models used in this study: U2OS (blue shading) and RCC (orange shading) together with Western blot. (I, J) POINT-seq signal on gene examples from class I (I) and class II (J), RCC SETD2 WT (red) and SETD2 mutation (orange). (K, L) POINT-seq metagene profile for class I (K) and class II (L) genes. (M) Termination index calculated for class I (left panel) and class II (right panel, brighter colors and gray shading) genes in RCC SETD2 WT (red) and SETD2 mutation (orange) cells. Data information: Boxplots on (G) and (M) show the median (center line), first and third quartiles (lower and upper bounds of the box), and whiskers extending to the most extreme data points within 1.5×IQR of the lower and upper quartiles, Mann–Whitney test was applied, ****$p \leq 0.0001$ (exact $p$-values for all figures are provided in the Table EV1); metaplots in (C) and (D) are based on the average from three U2OS biological replicates; metaplots in (E) and (F) are based on the average from two U2OS biological replicates; metaplots in (K) and (L) are based on the average from two RCC biological replicates. Number of genes analyzed ($n$) for U2OS class I: 4370 (C, E, G); U2OS class II: 1399 (D, F, G); RCC class I: 4892 (K, M); RCC class II: 867 (L, M). Source data are available online for this figure.

white shading) and class II genes (Fig. 1G, gray shading), even though the increase was notably bigger for class II genes. Interestingly, this analysis revealed additionally that already in WT conditions transcription termination of class II genes is more 'leaky' compared to class I genes, as class II genes have a higher termination index (Fig. 1G, black vs gray graphs).

Following this, we checked for other features characterizing class II genes compared to class I genes in WT cells—i.e., features that could be predictive of a gene's response to SETD2 loss. We found that class II genes have higher levels of active histone marks (Fig. EV1E–H), lower levels of repressive histone marks (Fig. EV1I,J), higher RNA expression (Fig. EV1K), higher elongation rate (Fig. EV1L), are longer (Fig. EV1M), and harbor longer introns (Fig. EV1N).

To summarize, our analysis indicates that SETD2 contributes to gene definition in several ways. First, by stimulating transcription initiation for class I genes (Fig. 1A,C). Second, class II genes (which tend to be longer, more highly expressed, and terminate later than the genomic average already in WT conditions) do not require SETD2 for normal transcription initiation but instead SETD2 prevents transcription readthrough downstream of those genes. Third, transcription termination is generally perturbed upon SETD2 loss. We predict therefore that—from the transcriptomics perspective—SETD2 loss can have detrimental effects by two mechanisms. First, by decreasing transcriptional output of class I genes which rely on SETD2 for efficient transcription initiation. Second, class II genes instead undergo transcription readthrough, which could result in transcriptional interference and/or delay RNAPII recycling for subsequent rounds of transcription. The observed SETD2-depletion induced readthrough phenotype of class II genes appears to be the result of their SETD2-independent initiation competence combined with a higher dependence on SETD2 for transcription termination compared to class I genes.

## Transcriptional effects of SETD2 loss are mediated by its methyltransferase activity

To address the question if the transcriptional effects of SETD2 uncovered here are dependent on its methyltransferase activity, and to corroborate our findings in a more biomedically relevant system, we moved to a renal cell carcinoma (RCC) model. *SETD2* mutations occur in approximately 13–20% in the specific subtype of renal cancers—clear cell renal cell carcinoma (ccRCC). The most frequent type of mutations are shallow deletions that, in

consequence, lead to reduced *SETD2* expression (Fig. EV1O,P). Loss of H3K36me3 creates a permissive epigenetic landscape promoting metastasis through enhanced chromatin accessibility and activation of oncogenic enhancers (Xie et al, 2022). We therefore employed cell lines derived from two RCC patients: ACHN with wild-type SETD2 (WT) and A498 with SETD2 mutation (mut) disrupting its catalytic activity (Fig. 1H); and performed POINT-seq (Figs. 1I–L and EV1Q,R). Our analysis in RCC cells recapitulated for a large part the observations made in SETD2 KO cells. Similarly to SETD2 KO U2OS cells, only a fraction of genes (15%) showed transcriptional readthrough in SETD2 mut RCC cells and was classified as class II (Fig. 1J,L). 85% of genes (class I) had instead strongly decreased transcription initiation (Fig. 1I,K). A difference from U2OS cells is that transcription initiation in RCC cells seems even more dependent on SETD2. Initiation of most genes was down about by a third in U2OS cells (Fig. 1C), while by approximately 3/4 in RCC (Fig. 1K). For class II genes, initiation was hardly affected in U2OS cells (Fig. 1D), while down to about half in RCC (Fig. 1L). We need to stress here the limitation of our approach: the two RCC cell lines are derived from two different patients and therefore diverse both genetically and cell-type-wise. In contrast, U2OS WT and KO cells share a common background. As a result, the magnitude of the effects observed in the RCC and U2OS systems cannot be directly compared. The overlap between class II genes in the RCC and U2OS models was only partial, but statistically significant (pval = 0.00054) (Fig. EV1S). Also, class II genes in the RCC model were characterized by a higher expression level (Fig. EV1T), increased length (Fig. EV1U–V) and higher termination index (Fig. 1M) already in the control situation, consistently with the findings in U2OS. We interpret the limited overlap between the models to be a result of osteosarcoma and RCC cells having different transcriptional profiles, and class II genes being associated with higher expression levels.

We conclude that the transcriptional effects of SETD2 loss can be likely attributed to its methyltransferase activity. Further on, those effects appear consistent between osteosarcoma and renal cell carcinoma cell lines—implicating a general, cell origin-independent mechanism.

## SETD2 protein and methyltransferase activity suppress cryptic transcription initiation

We noticed that the POINT-seq slopes after SETD2 depletion or catalytic mutation changed, especially for class II genes (Fig. 1D,L).

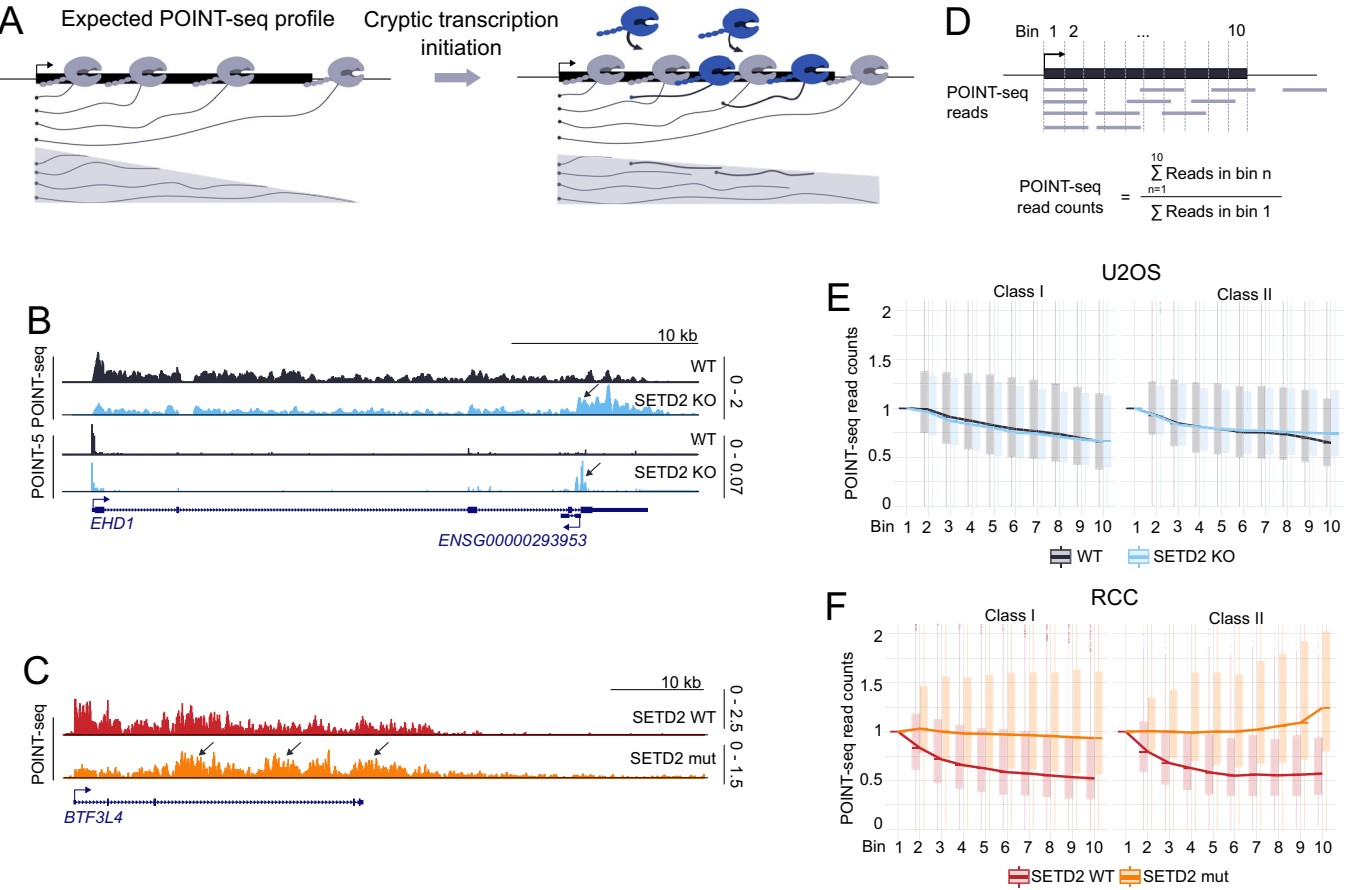

**Figure 2. SETD2 protein and its methyltransferase activity suppresses cryptic transcription initiation.**

(A) Schematic representation of theoretical POINT-seq profile in normal conditions (left panel) and upon cryptic transcription initiation (right panel). RNAPII initiating in the gene body are colored in dark blue. (B) POINT-seq and POINT-5 signal on a gene example from class II, in U2OS WT (black) and SETD2 KO (blue). Cryptic initiation is indicated by arrows. (C) POINT-seq signal on a gene example from class II, RCC SETD2 WT (red) and SETD2 mutation (orange). Cryptic initiation is indicated by arrows. (D) Diagram illustrating calculation of normalized POINT-seq read counts in genic bins performed for analysis in (E) and (F). First, analyzed genes were divided into 10 bins of equal length. Next, the number of reads in each bin was summed, and the resulting values were normalized by dividing by the number of reads in the first bin. (E) Boxplot representing the number of normalized POINT-seq read counts for class I, $n = 4229$ (left) and class II, $n = 1359$ (right) genes in U2OS WT (black) and SETD2 KO (blue). (F) Boxplot representing the number of normalized POINT-seq read counts for class I, $n = 4703$ (left) and class II, $n = 862$ (right) genes in RCC SETD2 WT (red) and SETD2 mutation (orange). Data information: Boxplots on (E) and (F) show the median (center line), first and third quartiles (lower and upper bounds of the box), and whiskers extending to the most extreme data points within 1.5×IQR of the lower and upper quartiles, (E) is based on the average from three U2OS biological replicate and (F) is based on the average from two RCC biological replicates.

We found that this change in slope is theoretically consistent with increased cryptic initiation within the gene body (Fig. 2A), a process previously suggested to be inhibited by SETD2/H3K36me3 (Carvalho et al, 2013). We were able to detect cryptic initiation events upon SETD2 KO/mut examining our POINT-seq datasets on individual genes (Fig. 2B upper panel, Figs. 2C and EV2C,D upper panel, EV2E). To globally corroborate the incidence of cryptic initiation using our POINT-seq data, we calculated the read count per bin in 10 equal genic bins for class I and II genes, and normalized it to the first, TSS-proximal bin (Fig. 2D). This allowed us to compare the amount of nascent RNA in the gene body, to its level at normal initiation sites. In control conditions (SETD2 WT) the normalized nascent RNA signal dropped in each bin as compared to the previous one (Fig. 2E black line, Fig. 2F red line), according to the theoretical model in Fig. 2A. This was true both for class I and II genes (except for last bins of class II genes in RCC, discussed more below). However, for class II genes

both in SETD2 KO and mutant conditions, an increase of normalized nascent RNA signal could be observed in the gene body, as compared to WT condition (Fig. 2E,F). This effect was particularly evident in the last bins (9 and 10), close to the PAS. This suggests that SETD2 can suppress cryptic initiation, especially in the vicinity of gene ends. SETD2 suppression of cryptic initiation was much stronger in RCC cells, which could be due to the cells lines being derived from different patients.

Since cryptic initiation appeared less pronounced in U2OS and our measurements by the indirect POINT-seq method were not entirely convincing for this cell line, we wanted to confirm them with a method that would allow us to directly measure 5' ends of nascent RNA. To this end, we performed POINT-5 in U2OS cells with ExoTerminator nuclease, a method as specific as CAGE (Sousa-Luís et al, 2021) (Fig. EV2A). POINT-5 analysis confirmed that the normalized read count of initiating

reads was elevated in U2OS SETD2 KO compared to WT, and that the difference was more prominent for class II genes, particularly for bins 9 and 10 (Fig. EV2B). This analysis additionally revealed that cryptic initiation frequency tends to drop from 5' to 3' end of genes, in both WT and SETD2 KO conditions (Fig. EV2B). We then estimated the frequency of cryptic initiation upon SETD2 KO, using a stringent approach relying on presence of both POINT-5 and POINT-seq evidence (see Methods). When probing the entire gene body, we observed the appearance of cryptic initiation in KO vs WT condition in about 20% of genes (18% for class I and 21.5% for class II genes; Fig. EV2F). When focusing on gene 3' ends (last 20% of the gene body), the frequency was 8.5% overall (5.8% for class I and 11.15% for class II; Fig. EV2G).

In conclusion, our data supports the claim that SETD2 suppresses cryptic initiation. This effect is dependent on SETD2 methyltransferase activity, occurs both in cell lines from osteosarcoma and renal origin, being more pronounced in the second case. Therefore, RCC cells appear more dependent on SETD2 activity for both normal initiation (Fig. 1K) and suppression of cryptic initiation (Fig. 2F, left panel) as compared to U2OS cells (Fig. 1C and Fig. 2E, left panel). However, since the RCC cell lines are derived from different patients, SETD2 activity might not be the only factor contributing to enhanced initiation. In both U2OS and RCC models, class II genes are more prone to cryptic initiation, compared to class I genes.

## SETD2 protein and methyltransferase activity enhance pre-mRNA 3' cleavage efficiency, but only for class II genes

So far, we mostly focused our attention on transcription initiation, while investigation of readthrough transcription requires especially analysis of 3' ends. To detect the exact sites of pre-mRNA 3' end cleavage and polyadenylation (PAS) in both our cellular models we performed nuclear 3'mRNA-seq (Fig. EV3A,B) and used it for PAS usage definition (see Methods). We employed POINT-seq raw reads to measure cleavage efficiency on the experimentally uncovered PAS in both models (Figs. 3A–C and EV3C). We looked for changes in the frequency of POINT-seq reads spanning the PAS, which are indicative of uncleaved nascent RNA. Since pre-mRNA 3' cleavage is intrinsically imprecise (Tian et al, 2005) we detected uncleaved RNA in a 10 bp window surrounding the PAS (5 bp upstream and downstream of the main cleavage site) for each analyzed gene. For normalization, we counted POINT-seq reads in the region 250 bp upstream of the PAS window (Fig. 3A). As exemplified in Fig. 3B,C, we were able to develop a cleavage index which differentiates between efficient cleavage (high cleavage index) and inefficient cleavage (lower cleavage index). This allowed us to perform global analysis of cleavage in U2OS (Fig. 3D) and RCC cells (Fig. 3E). We found that class I genes showed no significant difference between WT and SETD2 KO/mut conditions in either cell line. In contrast, cleavage index of class II genes dropped significantly between WT and KO/mut conditions, in both U2OS and RCC cells (Fig. 3D,E). This data suggests that SETD2 methyltransferase activity has an enhancing effect on pre-mRNA 3' cleavage, specifically on class II genes. Similarly to the readthrough phenotype (Fig. 1), this effect is general (occurring with similar frequency and strength in U2OS and RCC cells). To test the

relationship between the cleavage efficiency and the extent of readthrough transcription, we calculated the Spearman's rank correlation coefficient for termination and cleavage indexes (Fig. EV3D,E). We observed a significant (pval < 0.0001) negative correlation between the termination index and the cleavage index, for both WT and SETD2 KO/mutation conditions, and both gene classes. Therefore, higher termination index (corresponding to more readthrough) is associated with lower cleavage index, i.e., less efficient termination. We conclude that the enhancing effect of SETD2 methyltransferase activity on pre-mRNA 3' end cleavage of class II genes can likely contribute to SETD2 suppression of readthrough transcription on those genes.

## Transcriptional readthrough upon SETD2 activity loss is independent of alternative polyadenylation

Subsequently, we analyzed how the loss or enzymatic mutation of SETD2 influences alternative polyadenylation (APA). To this end, we performed APA analysis on our 3'mRNA-seq data (Fig. EV3A,B,F). Surprisingly, APA changes occurring in U2OS and RCC models upon SETD2 perturbation were entirely different. For U2OS, we found that genes harboring multiple alternative PASs were undergoing widespread APA in response to SETD2 KO, and exhibited predominantly a proximal shift (Fig. 3F, left bar chart, and Fig. 3G). This proximal shift was surprising, considering the readthrough phenotype of the KO, therefore we validated these results by RT-qPCR on a few representative genes (Fig. EV3G–I). Moreover, proximal and distal shifts were observed in both gene classes (Figs. 3F,G and EV3J). In contrast, in SETD2-mutated RCC cells, APA only occurred on a small subset of genes, with no preference towards distal/proximal direction (Fig. 3F right bar chart, Fig. EV3F). We conclude that the effect of SETD2 loss/mutation on APA is highly dependent on the cell origin. This contrasts with our previous assays measuring the effects of SETD2 on other aspects of transcription and RNA 3' processing, where SETD2 effects had the same direction and were similar in both cellular models (Figs. 1–3).

Our data suggests that APA and transcription readthrough upon SETD2 activity loss are independent of each other. There are three findings that support this conclusion. First, transcription readthrough upon SETD2 activity loss occurs in both cellular models on a subset of genes sharing common characteristics (higher transcriptional activity, higher termination index etc., Fig. EV1), while APA is predominant in U2OS but relatively rare in RCC. Second, the direction of transcription readthrough (distal) and APA (proximal) in U2OS are opposite (Fig. 3G). Third, we detect readthrough on genes undergoing APA in both directions, without APA change, and in single PAS genes (Figs. 3F,G and EV3F,J,K). Since we were not able to find any correlation between the changes in location of the PAS used and transcription readthrough, we conclude they are independent of each other.

## The effects of SETD2 activity on transcription are indirect

To assess whether the effects of SETD2 loss on transcription and pre-mRNA 3' processing are direct, we utilized a recently developed, specific inhibitor of SETD2 methyltransferase activity, EPZ-719 (Lampe et al, 2021). We treated U2OS cells with this

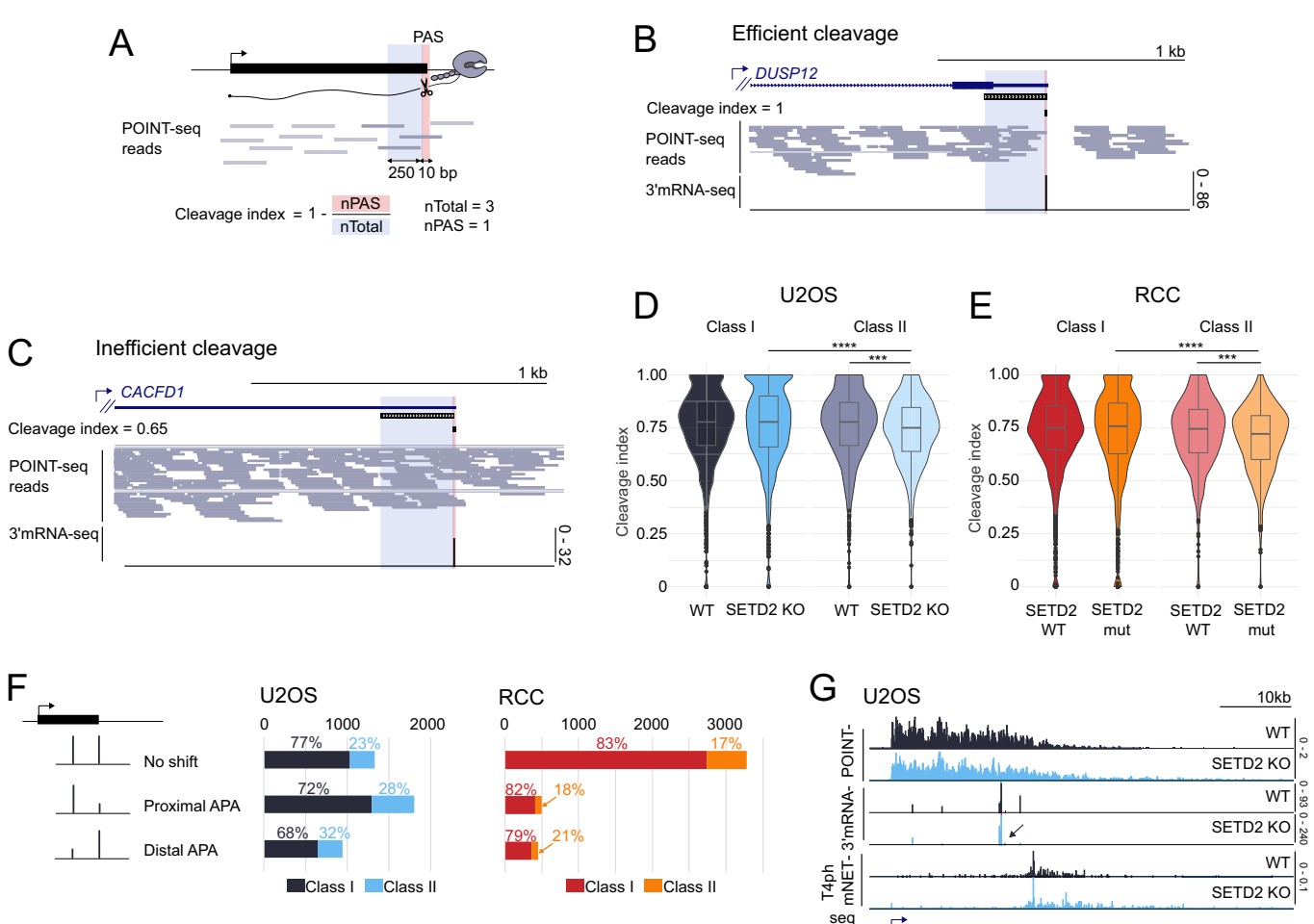

**Figure 3. SETD2 protein and its methyltransferase activity stimulate pre-mRNA 3′ end cleavage in class II genes; SETD2-induced readthrough is independent of APA.**

(A) Schematic representation of cleavage index calculation. For this analysis, only genes with one main PAS were selected. Then, two regions for each gene were established: (1) PAS window corresponding to 10 bp window with experimentally validated PAS in the middle and (2) total window of 250 bp length ending on PAS. POINT-seq reads were summed up in these windows and for each gene cleavage index was calculated as 1 − nPAS/nTotal. (B) Example of a gene with efficient cleavage, cleavage index = 1. (C) Example of a gene with inefficient cleavage, cleavage index = 0.65. (D) Cleavage index calculated for class I, $n = 1393$ (left) and class II, $n = 376$ (right) genes in U2OS WT (black, gray) and SETD2 KO (blue, lighter blue). (E) Cleavage index calculated for class I, $n = 1984$ (left) and class II, $n = 338$ (right) genes in RCC SETD2 WT (red, lighter red) and SETD2 mutation (orange, lighter orange). (F) Results of APA analysis depicted as bar charts with the distinction of class I and class II genes for U2OS (middle) and RCC (right). Multiple PAS genes were characterized either as "no shift" (no significantly changed PASs), "proximal APA" or "distal APA" depending on proximal-to-distal ratio between the two most significantly changed PASs. (G) POINT-seq, 3′mRNA-seq and T4ph mNET-seq signal on *SUB1* gene, which shows readthrough upon SETD2 loss (belongs to class II), yet undergoes proximal APA (indicated by an arrow). Data information: boxplots on (D) and (E) show the median (center line), first and third quartiles (lower and upper bounds of the box), and whiskers extending to the most extreme data points within 1.5×IQR of the lower and upper quartiles, Mann–Whitney test was applied, ***$p \leq 0.001$, ****$p \leq 0.0001$, cleavage index was calculated based on three or two biological replicates for (D) and (E), respectively; (F) is based on three biological replicates.

inhibitor and found that H3K36me3 levels decreased by 3 days of treatment (Fig. 4A). However, at this timepoint, little if any readthrough could be detected on representative class II genes in response to SETD2 methyltransferase activity inhibition (Fig. 4B–E). We only observed substantial transcription readthrough after 9 days of inhibition, and even then, its level did not reach the level observed in the SETD2 KO cells (Fig. 4C–E).

To assess the robustness of SETD2 activity on termination, we treated two non-cancerous cell lines (HEK293 and hTERT-RPE) with EPZ-719 (Fig. EV4A) and measured readthrough induction in several representative genes (Fig. EV4B–D). Interestingly, in hTERT-RPE cells the effect appeared earlier than in U2OS, with

clear readthrough induction after 3 days (72 h) of treatment (Fig. EV4B–D). However, a significant decrease in H3K36me3 levels was already observed after 24 h (Fig. EV4A), again indicating that the effect of SETD2 activity on termination is indirect.

Next, we asked whether the effect on termination is accompanied by a simultaneous decrease in initiation. We performed POINT-seq on U2OS cells treated with EPZ-719 for 9 days and found that readthrough occurred in a smaller subset of genes ($n = 493$), while initiation was not significantly affected in either class I or class II genes (Fig. 4F–I). We conclude that the effect of SETD2 methyltransferase activity on transcription readthrough is not caused by H3K36me3 loss directly. These results also suggest

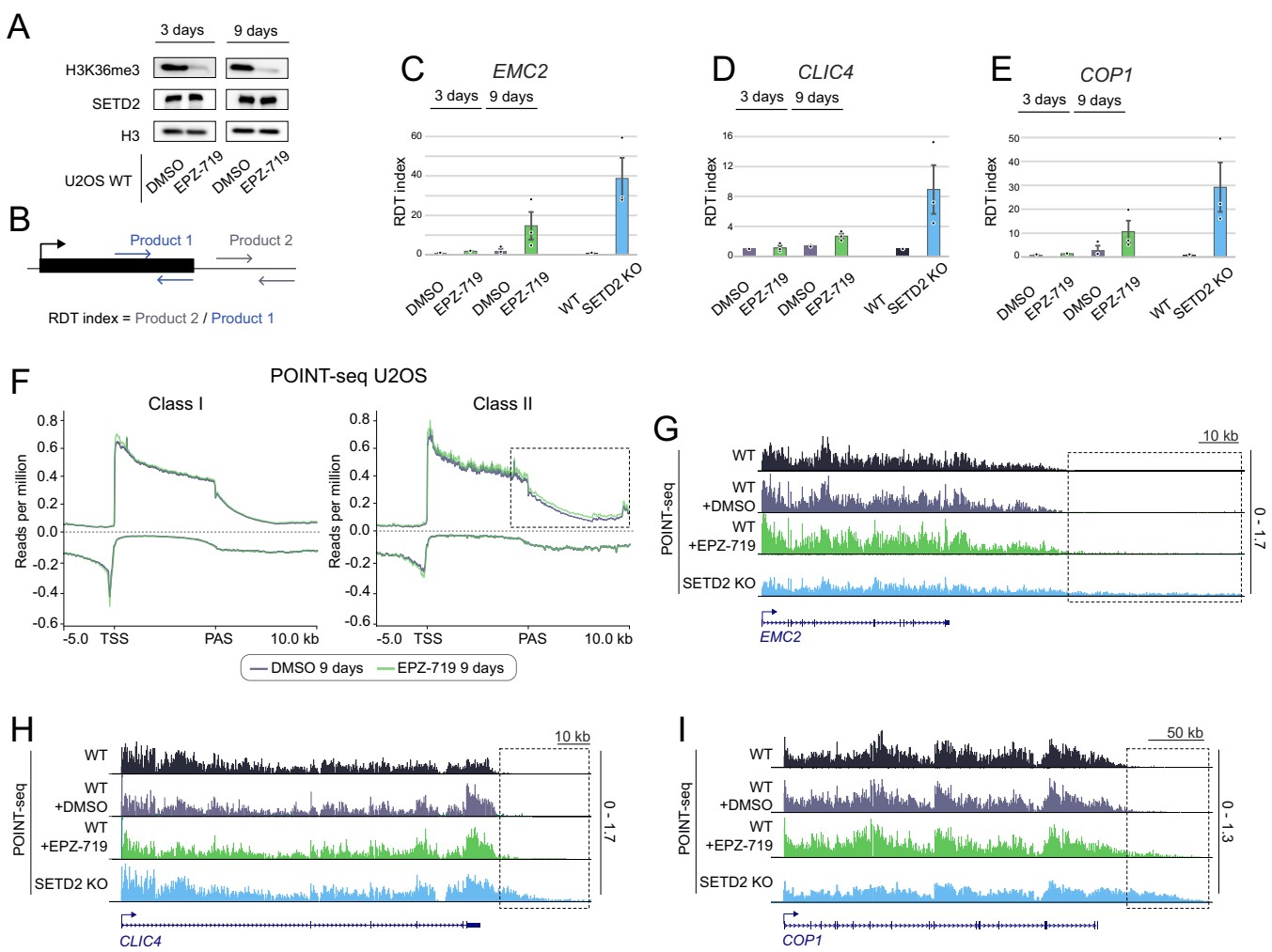

**Figure 4. The effects of SETD2 methyltransferase activity on transcription are indirect.**

(A) Western blot showing H3K36me3 and SETD2 abundance after treatment of U2OS WT cells with SETD2 inhibitor EPZ-719 after three days (left) and nine days (right). (B) Schematic representation of qPCR assay used for calculating readthrough (RDT) index. Primers were designed in intron (last for *EMC2* and *COP1*, first for *CLIC4*) (product 1) and downstream of PAS (product 2). RDT index was calculated by dividing the amount of product 2 by product 1. (C–E) Bar plot representing RDT index calculated for *EMC2*, *CLIC4* and *COP1* genes in U2OS WT cells (black) treated with DMSO (gray) or SETD2 inhibitor EPZ-719 (green) as well as for SETD2 KO (blue). (F) POINT-seq metagene profile of U2OS cells treated for 9 days with DMSO (gray) or SETD2 inhibitor EPZ-719 (green) for class I genes (left panel, $n = 5269$) and class II genes (right panel, $n = 493$). (G–I) POINT-seq signal on *EMC2*, *CLIC4* and *COP1* genes: U2OS WT (black), treated with DMSO (gray) or EPZ-719 (green and SETD2 KO (blue). Readthrough area is marked by a dotted rectangle. Data information: bars in (C–E) represent mean ± SEM; three biological replicates are represented as dots; (F) is based on the average from two biological replicates. Source data are available online for this figure.

that the termination defect occurs before impaired initiation, which appears to be a downstream consequence.

## SETD2 loss has a pleiotropic effect on transcription-related proteome

To identify the factor(s) responsible for transcription readthrough following SETD2 loss, we performed mass spectrometry (MS)-based proteome analyses of nuclear proteins in WT and SETD2 KO U2OS cells (Dataset EV1). Volcano plot analysis revealed that among de-regulated proteins, there was a tendency for protein downregulation (Fig. 5A). Interestingly, the most significant gene ontology categories of nuclear proteins downregulated upon SETD2 KO were related to RNA metabolism, transcription and

chromatin (Fig. EV5A). Therefore, we analyzed our proteomics dataset for proteins involved in the transcription cycle, RNA splicing and epigenetic processes. Indeed, we found that an overwhelming majority of affected proteins from those categories was downregulated upon SETD2 KO, and this included RNAPII itself (Figs. 5A and EV5B). We also examined the ratio of POINT-seq signal (SETD2 KO/WT) for genes encoding the most down-regulated transcription-related proteins. At the transcriptional level, these genes were either downregulated or showed read-through (Fig. EV5C). After 9 days of EPZ-719 treatment, we observed a similar pattern of transcriptional changes, although less pronounced (Fig. EV5D–E).

We conclude that SETD2 is globally required for proper functioning of transcription, RNA processing and epigenetics

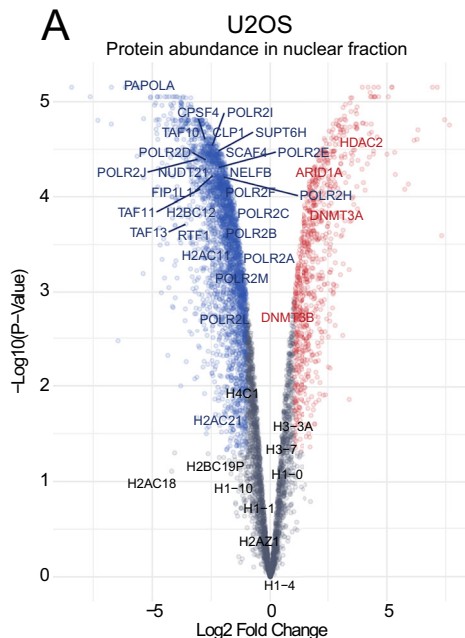

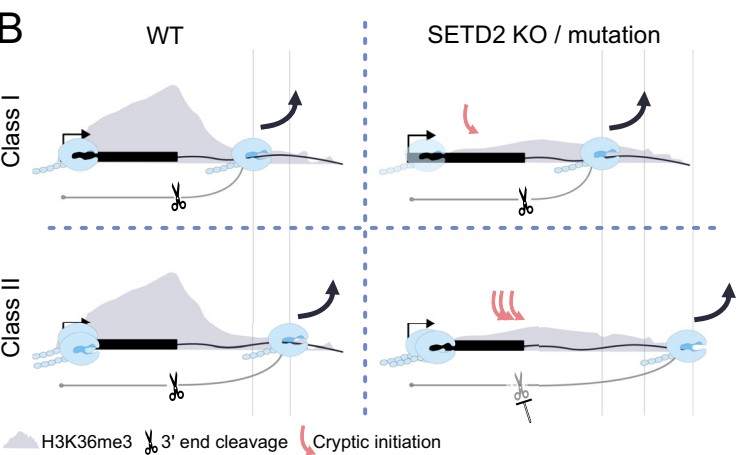

**Figure 5. SETD2 loss leads to downregulation of a plethora of transcription-related proteins; model of the influence of SETD2 on the two identified gene classes.**

(A) Volcano plot representing protein abundance in nuclear extract in U2OS SETD2 KO relative to SETD2 WT. Significantly downregulated and upregulated proteins are colored in blue and red, respectively. (B) Model illustrating the distinct roles of SETD2 in two gene classes. Upon SETD2 loss or mutation, TSS-linked transcription initiation is significantly disrupted in class I genes, whereas 3′ end cleavage and termination remain unchanged. In class II genes, SETD2 loss or mutation induces cryptic transcription initiation near gene ends, while TSS-linked initiation is not significantly affected. This cryptic initiation, together with decreased 3′ end cleavage efficiency, leads to readthrough transcription. Data information: (A) was based on three biological replicates; empirical Bayes-moderated t-test was applied.

pathways, and its loss leads to a global perturbation of the transcription-related proteome.

Our work demonstrates that SETD2 loss or catalytic inactivation disrupts both transcription initiation and termination in human cells, but in two different, gene-dependent ways. For most protein-coding genes, named here class I, SETD2 loss/inactivation predominantly perturbs TSS-linked transcription initiation, while pre-mRNA 3′ end cleavage remains largely unaffected, with no readthrough transcription. In contrast, a smaller subset of genes—class II—exhibits markedly different behavior. In the case of those genes, SETD2 loss/inactivation has little effect on initiation at the TSS, but leads to significantly disrupted 3′ end cleavage, and pronounced transcriptional readthrough. Additionally, cryptic initiation is substantially higher in class II compared to class I genes, both in SETD2 knockout and catalytic inactivation models (Fig. 5B). Both mechanisms occurring on class II genes in SETD2-deficient conditions—enhanced gene-body initiation and impaired 3′ end cleavage—can contribute to improper transcription termination resulting in readthrough transcription.

Class II genes are characterized by higher expression levels that are not drastically diminished upon SETD2 loss, compared to class I genes (Figs. 1D,L and EV1K,T), suggesting that elevated transcriptional activity may predispose these genes to readthrough and cryptic initiation. Furthermore, class II genes are significantly longer and harbor longer introns than class I genes (Fig. EV1M,N,U–V), pointing to a possible link between gene structure and susceptibility to SETD2-dependent transcriptional

regulation. Moreover, these genes are also transcribed faster in wild-type U2OS cells (Fig. EV1L), suggesting that they may depend on additional factors for efficient RNA processing. Crucially, SETD2's impact on transcription termination is independent of APA (Figs. 3F,G and EV3F–K), and the usage of long and short transcript isoforms (direction of APA) is similar between gene classes (Fig. 3F). Interestingly, class II genes can be differentiated already in unperturbed control condition from class I genes by less efficient termination, evidenced by higher termination index in WT conditions (Fig. 1G,M). Therefore, we predict that this 'leaky' gene class might be also more sensitive compared to most genes (class I) to perturbation of other epigenetic modifiers.

Using POINT-seq in RCC cells with catalytically inactive SETD2, and SETD2 enzymatic activity inhibition, we demonstrate that the methyltransferase activity of SETD2 is essential for maintaining termination fidelity across various cell types (Figs. 4 and EV4). However, this effect is indirect, requiring many days to build up, which points to an unidentified intermediate factor mediating this process. Moreover, the inefficient initiation observed in SETD2 KO seems to be a long-term consequence that follows defective termination. Reduced initiation levels may result from a diminished pool of available RNAPII molecules (Figs. 5A and EV5B) which remain engaged in the transcription of many genes for longer periods due to inefficient termination. Until now, the observed effects of long-term SETD2 depletion have been attributed in literature mainly directly to SETD2/H3K36me3. Our work shows that future studies are needed to discriminate between the direct

and indirect effects of SETD2/H3K36me3 on the many processes they have been implied in. This is now possible due to the development of a specific inhibitor, and various degron technologies.

# Methods

### Reagents and tools table

| Reagent/Resource | Reference or Source | Identifier or Catalog Number |
|---|---|---|
| **Experimental models** | | |
| U2OS | Timothy C. Humphrey Lab | |
| U2OS SETD2 CRISPR KO | Timothy C. Humphrey Lab | |
| ACHN | Purchased at ATCC | #CRL-1611 |
| A498 | Purchased at ATCC | #HTB-44 |
| HEK293 | JCRB Cell Bank | #2501067 |
| hTERT-RPE | Atsushi Shibata Lab | |
| **Recombinant DNA** | | |
| NA | | |
| **Antibodies** | | |
| Rabbit anti-SETD2 | Cell Signaling Technology | #80290 |
| Rabbit anti-H3 | Abcam | #ab1791 |
| Rabbit anti-H3K36me3 | Cell Signaling Technology | #9763 |
| Goat Anti-Mouse IgG H&L | Abcam | #ab205719 |
| Goat Anti-Rabbit IgG H&L | Abcam | #ab205718 |
| Mouse anti-RNAPII antibody | Hiroshi Kimura Lab, for POINT-seq | #MABI601 |
| Rat anti-RNAPII CTD T4ph antibody | Active Motif | #61361 |
| **Oligonucleotides and other sequence-based reagents** | | |
| PCR primers | This study | Table EV2 |
| **Chemicals, Enzymes and other reagents** | | |
| Dulbecco's Modified Eagle Medium (DMEM - High glucose) | Nacalai Tesque | Cat# 29113-53 |
| Fetal Bovine Serum (FBS) | Gibco | Cat# 10270-106 |
| Penicillin-Streptomycin Mixed Solution | Nacalai Tesque | Cat# 26253-84 |
| PBS | Nacalai Tesque | Cat# 14249-24 |
| Dimethyl sulfoxide (DMSO) | Sigma | Cat# D8418 |
| 0.5g/l-Trypsin/ 0.53 mmol/l-EDTA Solution | Nacalai Tesque | Cat# 35553-74 |
| NaCl | Nacalai Tesque | Cat# 31320-05 |
| MgCl$_2$ | Thermo Fisher Scientific | Cat# AM9530G |
| Sodium Dodecyl Sulfate (SDS) | Nacalai Tesque | Cat# 02873-75 |

| Reagent/Resource | Reference or Source | Identifier or Catalog Number |
|---|---|---|
| NP-40 | Sigma | Cat# 56741-250ML-F |
| Protease inhibitor | Nacalai Tesque (POINT-seq) MERCK (mNET-seq) | Cat# 03969-21 Cat# 04693159001 |
| Phosphatase inhibitor | Roche | Cat# 04906837001 |
| Benzonase | Millipore | Cat# E1014 |
| Microccal Nuclease Reaction Buffer (Mnase buffer) | NEB | Cat# M0247S |
| Micrococcal Nuclease (MNase) | NEB | Cat# M0247S |
| 2-Mercaptoethanol (2-ME) | Nacalai Tesque | Cat# 21438-82 |
| NuPAGE LDS buffer (4x) | Thermo Fisher Scientific | Cat# NP0007 |
| Immobilon-P Transfer Membrane (PVDF membrane) | Millipore | Cat# IPVH00010 |
| Tris-Glycine buffer(10X) (pH 8.3) | Nacalai Tesque | Cat# 09422-81 |
| Methanol | Wako | Cat# 139-01827 |
| PonceauS | Nacalai Tesque | Cat# 28322-72 |
| Skim Milk for immunoassay | Nacalai Tesque | Cat# 31149-75 |
| Tween-20 | Santa Cruz | Cat# Sc-29113 |
| TBS | Nacalai Tesque | Cat# 050.0 |
| Bovine Serum Albumin | Thermo Fisher Scientific | Cat#30063-481 |
| Chemi-Lumi One Super | Nacalai Tesque | Cat# 02230-30 |
| Sucrose | Nacalai Tesque | Cat# 30406-25 |
| 0.5 M EDTA (pH 8.0) | Nacalai Tesque | Cat# 06894-14 |
| Glycerol | Nacalai Tesque | Cat# 17045-65 |
| 1 M HEPES-KOH (pH 7.5) | Nacalai Tesque | Cat# 15639-84 |
| Urea | Nacalai Tesque | Cat# 35905-35 |
| Empigen BB (30%) | Sigma | Cat# 30326 |
| RiboLock (RNase inhibitor) | Thermo Fisher Scientific | Cat# EO0381 |
| Turbo DNase | Thermo Fisher Scientific | Cat# AM2239 |
| RNase H | NEB | Cat #M0297S |
| Dynabeads M280 Sheep Anti-mouse IgG | Thermo Fisher Scientific | Cat# 11202D |
| TRI Reagent | Cosmo Bio | Cat# TR118 |
| Chloroform | Sigma | Cat# C2432-500ML |
| Glyco Blue | Ambion | Cat# 1405030 |

| Reagent/Resource | Reference or Source | Identifier or Catalog Number |
|---|---|---|
| 2-Propanol | Nacalai Tesque | Cat# 29113-53 |
| 10X Turbo DNase buffer | Invitrogen | Cat# AM2238 |
| Ethanol | Nacalai Tesque | Cat# 14713-53 |
| Trypan Blue stain 0.4% | Invitrogen | Cat# T10282 |
| SPRISelect reagent | Beckman Coulter | Cat# B23317 |
| Mini-PROTEAN TGX Precast Gels 4-20% | Bio Rad | Cat# 456-1096 |
| Running Buffer Solution(10X) for SDS-PAGE | Nacalai Tesque | Cat# 30329-61 |
| WIDE RANGE Gel Preparation Buffer(x4) for PAGE | Nacalai Tesque | Cat# 07831-94 |
| 40(w/v)%-Acrylamide/Bis Mixed Solution (37.5:1) | Nacalai Tesque | Cat# 06121-95 |
| Ammonium Peroxodisulfate (APS) | Nacalai Tesque | Cat# 02602-02 |
| TEMED | Nacalai Tesque | Cat# 33401-72 |
| Precision Plus Protein Dual Colour Standards | Bio Rad | Cat# 1610374 |
| UltraPure™ SSC, 20X | ThermoFisher | Cat# 15557044 |
| Dextran Sulfate Sodium | Nacalai Tesque | Cat# 03879-72 |
| Triton X-100, Molecular Biology Grade | MERCK | Cat# T8787-100ML |
| 0.5g/l-Trypsin/ 0.53 mmol/l-EDTA Solution | Nacalai Tesque | Cat# 35553-74 |
| EPZ-719 | Medchemexpress | #2697176-16-0 |
| NEBNext Ultra II Directional RNA library prep kit for Illumina | NEB | Cat# E7760S1 |
| Direct-zol RNA Microprep kit | Zymo Research | Cat# R2062 |
| Protein G Dynabeads | Thermo Fisher Scientific | Cat# 10004D |
| EGTA | MERCK | Cat# E3889 |
| 1x T4 PNK buffer | NEB | Cat# M0236S |
| ATP | Cell Signaling | Cat# 9804S |
| NEBNext Small RNA Library Prep Set for Illumina | NEB | Cat# E7330L |
| QuantSeq 3′ mRNA-seq Library Prep Kit REV | Lexogen | Cat# 016.96 |
| **Software** | | |
| FastQC v0.12.1 | https://www.bioinformatics.babraham.ac.uk/projects/fastqc/ | |
| TrimGalore v0.6.6 | https://www.bioinformatics.babraham.ac.uk/projects/trim_galore/ | |
| STAR v2.7.6a | Dobin et al, 2013 | |

| Reagent/Resource | Reference or Source | Identifier or Catalog Number |
|---|---|---|
| SAMtools v1.10 | Li et al, 2009 | |
| Bedtools v2.31.1 | Quinlan and Hall, 2010 | |
| bedGraphToBigWig | Perez et al, 2025 | |
| WiggleTools v1.2.11 | Zerbino et al, 2014 | |
| MACS2 | Zhang et al, 2008 | |
| deepTools v3.5.5 | Ramírez et al, 2016 | |
| R v.4.2.0-4.5.1 | https://www.r-project.org | |
| DIA-NN v2.0.2 | https://aptila.bio | |
| DESeq2 v1.50.2 | Love et al, 2014 | |
| BRGenomics v1.12.0 | https://mdeber.github.io | |
| rtracklayer v1.54.0 | Lawrence et al, 2009 | |
| plyranges v1.14.0 | Lee et al, 2019 | |
| DEXSeq v1.56.0 | Anders et al, 2012 | |
| **Other** | | |
| ChemiDoc Touch Imaging System | Bio Rad | Cat# 17001401JA |
| Thermo Mixer C | Eppendorf | Cat# 5382000023 |
| NovaSeq6000 | Illumina | Novogene |
| Qubit 4 Fluorometer | Thermo Fisher Scientific | Cat# Q33238 |

## Experimental material and methods

### Cell culture

Details of U2OS cells (U2OS parental cell, U2OS SETD2 KO cell) were previously published (Pfister et al, 2014). U2OS cells were cultured in high glucose Dulbecco's modified Eagle's medium (DMEM) with 10% fetal bovine serum (FBS) and penicillin/streptomycin. ACHN, A498, HEK293, and hTERT-RPE cells were cultured in Eagle's minimum essential medium (EMEM) with 10% fetal bovine serum (FBS) and penicillin/streptomycin. All the cells were maintained in a humidified incubator at 5% $CO_2$, 37 °C. Cells were tested for mycoplasma contamination using PCR Mycoplasma Detection Set (Takara, #6601) and no mycoplasma was detected.

### Western blot

$1.0 \times 10^4$ cells were treated with lysis buffer (50 mM Tris-HCl pH 7.5, 150 mM NaCl, 2 mM $MgCl_2$, 1% NP-40, 5% Glycerol, 0.2% SDS, 1X Protease inhibitor, ~10 units Benzonase) to digest DNA and RNA, then incubated for 10 min at 37 °C. Then LDS buffer was added to the solution and incubated for 10 min at 70 °C. The primary and secondary antibodies were diluted as indicated below with skim milk in TBS-T buffer. The primary and secondary antibodies used in this study are listed below. Chemi-Lumi One Super was used for chemiluminescence. Images were captured by the Chemidoc Touch V3 system (Bio-Rad).

### Antibodies used in western blot

SETD2 (rabbit, Cell Signaling Technology, #80290) 1:1000, H3 (rabbit, Abcam, #ab1791) 1:50,000, H3K36me3 (rabbit, Cell

Signaling Technology, #9763) 1:3000, Goat Anti-Mouse IgG H&L (Abcam, #ab205719) 1:30,000, Goat Anti-Rabbit IgG H&L (Abcam, #ab205718) 1:10,000.

### SETD2 inhibitor treatment

EPZ-719 was dissolved in DMSO and directly added to U2OS/HEK293/hTERT-RPE cells in DMEM/EMEM with 10% FBS/penicillin/streptomycin (final concentration 1 μM). For long-term treatment, cells were passaged every 3 days and EPZ-719 was added freshly every time.

### RT-qPCR

Total RNA was purified from U2OS or U2OS SETD2 KO cells with Direct-zol kit (ZYMO). According to the manufactual protocol, 500 ng of total RNA was used with Superscript III kit (Invitrogen) and N6 random primers (Invitrogen) for cDNA synthesis. To ensure reproducibility, RNA purification and cDNA synthesis were performed on the same day. The cDNA was amplified with indicated primers and KAPA SYBR Fast qPCR kit (Roche) for quantitative real-time PCR (LightCycler 96, Roche). The experiments were repeated three times for three biological replicates. For absolute quantification of qPCR signals, genomic DNA of U2OS or U2OS SETD2 KO cells were serially diluted. Primer pairs were designed for specific regions downstream of the annotated polyadenylation site and for an intronic region of the same transcript. The readthrough index was calculated as the ratio of the downstream amplicon to the intronic amplicon ($2^{-\Delta Ct\_downstream}/2^{-\Delta Ct\_intron}$). For APA validation, RNA for 3'mRNA-seq libraries preparation was used. Primers were designed to target the 3rd exon (total) and 3'UTR (distal isoform). Next, ratio distal/total was calculated based on the starting RNA quantity (ng) obtained based on the standard curve.

Primer sequences for RT-qPCR can be found in Table EV2.

### Polymerase intact nascent transcript sequencing (POINT-seq)

POINT-seq was performed as previously described (Sousa-Luís et al, 2021) with minor modifications. Typically, 10 million cells were lysed with 4 mL ice-cold hypotonic buffer (10 mM Tris-Cl pH 7.5, 10 mM NaCl, 2.5 mM MgCl$_2$, 0.5% NP-40) on ice for 5 min. The cell lysate was carefully underlaid with 1 mL sucrose cushion buffer (hypotonic buffer+10% (w/v) sucrose). The nuclear fraction was pelleted by centrifugation ($300 \times g$, 5 min at 4 °C), then washed with ice-cold hypotonic buffer (without NP-40) once. The purified nuclear fraction was first resuspended in 300 μL ice-chilled NUN1 buffer (20 mM Tris-HCl pH 7.9, 75 mM NaCl, 0.5 mM EDTA pH 8.0, and 50% glycerol) and then lysed in 3 mL (at least 10-fold volume of NUN1) ice-cold NUN2E buffer (20 mM HEPES-KOH pH 7.6, 300 mM NaCl, 0.2 mM EDTA pH 8.0, 7.5 mM MgCl$_2$, 1% NP-40, 1 M Urea, 3% Empigen BB, 1xProtease inhibitor Complete EDTA-free, and 1xPhosSTOP). Immediately after adding NUN2E, chromatin glue was visible. The chromatin was washed with 10 mL ice-cold PBS once and then digested in DNase mixture (10 mM Tris-HCl pH 7.5, 400 mM NaCl, 10 mM MnCl$_2$, 2 Units/ μL RiboLock, and 0.2 Units/μL Turbo DNase) at 37 °C for 15 min at 1200 rpm in the ThemoMixer C (Eppendorf). EDTA pH 8.0 (final 1 mM) was used to stop the DNase reaction, and the supernatant was collected by high-speed centrifugation ($14,000 \times g$, 10 min at 4 °C). A five-fold volume of ice-cold NET-2E buffer (50 mM Tris-HCl pH 7.4, 150 mM NaCl, 0.05% NP-40, and 3% Empigen BB)

was used for the dilution of the supernatant. The 10 μg of anti-RNAPII antibody MABI601 were conjugated with 200 μL anti-mouse IgG Dynabeads solution in 1 mL NET-2 buffer and incubated at 4 °C for 1 h. The conjugated beads were washed with NET-2 buffer once and added to the 5-fold diluted supernatant and then incubated at 4 °C for 1 h. After the incubation, beads were washed with 1 mL of ice-cold NET-2E buffer at least six times. RNAPII intact nascent transcripts (POINTs) fraction were treated with Turbo DNase once at 37 °C for 15 min and then purified using Direct-zol kit following the manufacturer's protocol. The POINTs were fragmented to 150–300 nucleotides for sequencing according to the protocol of NEBNext Ultra II Directional RNA library prep kit. The POINT-seq libraries were sequenced in Illumina PE150 NovaSeq6000 platform (Novogene).

### POINT-5

RNAPII intact nascent transcript (POINT) was obtained as described above. 300–500 ng of POINT was then treated with Terminator 5'-Phosphate-Dependent Exonuclease using buffer A (BIOSEARCH TECH, #TER51020) for 60 min at 30 °C in thermomixer at 1000 rpm to degrade 5'p-RNAs which are mainly made by co-transcriptional RNA cleavage. The undegraded RNAs were purified using Direct-zol kit by following the manufacturer's protocol. The templated switching approach with N6 random primers was applied to prepare POINT-5 library by following the protocol of the SMARTer Stranded RNA-seq kit. The size-selected POINT-5 libraries (200–800 bp) were sequenced in Illumina PE150 NovaSeq6000 platform (Novogene).

### T4ph mNET-seq

For each sample, 50 μL of protein G Dynabeads (Thermo Fisher Scientific, 10004D) were prepared for immunoprecipitation of T4ph RNAPII (Active Motif, 61361). The beads were washed twice with ice-cold NET2+Empigen buffer (50 mM Tris pH 7.4, 150 mM NaCl, 0.05% NP-40, 1% Empigen BB), incubated overnight at 4 °C with 5 μg of T4ph antibody, washed twice with ice-cold NET2 buffer, and kept on ice until use.

Cells grown on 150 mm dishes were rinsed with ice-cold PBS and collected by scraping on ice. The cell suspension was transferred to 10 mL NUNC tubes and centrifuged at $500 \times g$ for 5 min at 4 °C. Pelleted cells were resuspended in 4 mL of ice-cold HLB + N buffer (10 mM Tris pH 7.5, 10 mM NaCl, 2.5 mM MgCl$_2$, 0.5% NP-40) and lysed on ice for 6 min. Lysates were underlaid with 1 mL of ice-cold HLB + NS buffer (same as HLB + N, but with 10% sucrose) and centrifuged at $500 \times g$ for 5 min at 4 °C. Nuclei were resuspended in 120 μL of ice-cold NUN1 buffer (20 mM Tris pH 7.9, 75 mM NaCl, 0.5 mM EDTA, 50% glycerol). The nuclear suspension was then transferred to a fresh tube, mixed with 1.2 mL of ice-cold NUN2 buffer (20 mM HEPES-KOH pH 7.6, 300 mM NaCl, 0.2 mM EDTA, 7.5 mM MgCl$_2$, 1% NP-40, 1 M urea, PhosSTOP [MERCK, 4906837001], and protease inhibitors [MERCK, 04693159001]), vortexed at maximum speed, and incubated on ice for 10 min with periodic vortexing every 3–4 min.

Chromatin was pelleted by centrifugation at $5000 \times g$ for 1 min at 4 °C, then washed with 500 μL ice-cold PBS and 100 μL ice-cold 1x MNase buffer (NEB, M0247). Chromatin was digested in 100 μL MNase reaction buffer containing 40 units/μL MNase at 37 °C for 2–5 min in a thermomixer (1200 rpm). Digestion was stopped by adding 12.5 μL of 0.2 M EGTA, and the mixture was centrifuged at

16,000 × g for 5 min at 4 °C. The resulting 100 µL supernatant was diluted tenfold with 900 µL ice-cold NET2+Empigen buffer, and the antibody-conjugated beads were added for immunoprecipitation at 4 °C for 1 h.

Beads were washed six times with 1 mL ice-cold NET2+Empigen buffer and once with 50 µL ice-cold PNKT buffer (1x T4 PNK buffer [NEB, M0236S], 0.1% Tween 20). The beads were then incubated in 50 µL PNK reaction mix (1x PNKT, 1 mM ATP, 0.05 U/mL T4 PNK 3' phosphatase minus [NEB, M0236S]) at 37 °C for 5 min in a thermomixer (1200 rpm), followed by a wash with NET2+Empigen buffer.

RNA was extracted from the beads using 1 mL TRI Reagent Solution (ThermoFisher, AM9738), followed by chloroform extraction and isopropanol precipitation. The RNA pellet was resuspended in 10 µL urea dye (7 M urea, 0.05% xylene cyanol, 0.05% bromophenol blue), denatured, and separated on a 6% TBE-Urea gel (Thermo Fisher Scientific, EC68652BOX) at 180 V for 15–18 min. RNA fragments between 25–100 nt were size-selected by excising the gel region between the bromophenol blue and xylene cyanol markers. Gel pieces were disrupted by centrifugation through a pierced 0.5 mL tube nested in a 1.5 mL tube at 16,000 × g for 1 min. RNA was eluted from the gel using RNA elution buffer (1 M NaOAc, 1 mM EDTA) at 25 °C for 1 h in a thermomixer (1200 rpm). The eluate was passed through a SpinX column (Costar, Corning, 8160) containing two glass filters (Whatman, WHA1823010), and the RNA was ethanol precipitated.

mNET-seq libraries were constructed using the NEBNext Small RNA Library Prep Set for Illumina (NEB, E7330L) and amplified with 15 PCR cycles. Libraries were size-selected on a 6% TBE gel (Thermo Fisher Scientific, EC62652BOX), isolating 150–230 bp PCR products to remove primer dimers. Gel elution was performed as described above. Sequencing was carried out on NovaSeq 6000 platform. Two biological replicates were processed for each cell line.

### 3'mRNA-seq

Nuclei were isolated following the same procedure as described for mNET-seq. Nuclear pellets were washed with 1 mL of ice-cold PBS, then resuspended in 1 mL of TRI Reagent™ Solution (Thermo Fisher Scientific, AM9738). RNA extraction and ethanol precipitation were carried out according to the manufacturer's instructions. To remove genomic DNA, samples were treated with 4 units of TURBO DNase (Thermo Fisher Scientific, AM2238) at 37 °C for 30 min, followed by a second round of TRI Reagent™ Solution purification. RNA integrity was assessed using an Agilent Tapestation. For library preparation, 500 ng of total RNA was used as input for the Lexogen QuantSeq 3' mRNA-Seq Library Prep Kit REV, following the manufacturer's protocol. Libraries were amplified with 12 PCR cycles, and their size distribution was checked on an Agilent Tapestation. Indexed libraries were quantified, pooled, and sequenced on an Illumina NovaSeq 6000. Three biological replicates were processed for each condition.

### Proteome analysis

Proteins were extracted from pellet samples of 10 million cells with 100 µL 8 M urea, 100 mM Tris-HCl pH 8.5 buffer containing 0.02% lauryl maltose neopentyl glycol (LMNG), 5 mM tris(2-carboxyethyl) phosphine hydrochloride (TCEP), 30 mM 2-chloroacetamide (CAA) and benzonase. 50 mM ammonium bicarbonate (ABC) solution was added to decrease the urea concentration to 5 M. Then, mass

spectrometry grade lysyl endopeptidase (Lys-C) (Wako) was added at 1:50 w/w and the samples were incubated at 37 °C for one hour. 50 mM ABC was added to decrease the urea concentration to 1.5 M. Sequencing grade modified trypsin (Promega) was added at 1:50 w/w and the samples were incubated at 37 °C for about 16 h. Afterwards, trifluoroacetic acid (TFA) was added to 0.7% (v/v). Tryptic peptides were desalted with Evotip Pure tips (Evosep), ready for mass spectrometry measurement.

For mass spectrometry, an Evosep One LC system coupled to an Orbitrap Exploris 480 mass spectrometer (Thermo Fisher Scientific) was used. Measurements were made with data-independent acquisition (DIA). Peptides were separated on a 75-µm ID analytical column filled with 1.9-µm C18 particles to 15-cm filling length (Bruker), at 35 °C in a column oven (AMR, Inc.). The column was attached to a 10-µm emitter tip (Evosep). Mobile phases A and B were 0.1% formic acid and 0.1% formic acid in 100% acetonitrile, respectively. A proprietary 58-minute LC gradient method was used "methXcalibur Whisper100 20 SPD (58 min, IonOpticks Aurora Elite, EV1112).cam" (Evosep). The following settings were maintained throughout data acquisition: positive mode; electrospray voltage, 2.0 kV; ion transfer tube temperature, 275 °C; default charge state, 3; orbitrap resolution, 60,000; RF lens, 40%; centroid data. The MS cycles were: One full MS scan from 495–745 $m/z$, followed with 40 DIA scans with a precursor mass range of 498–742 $m/z$ and an isolation window of 6 $m/z$; then another full MS scan from 495–745 $m/z$, followed by 39 DIA scans with a precursor mass range of 500–740 $m/z$ and an isolation window of 6 $m/z$. For full MS scans, the normalized AGC target was 300%. For DIA scans, normalized HCD collision energies were 22, 26, and 30%, scan ranges were from 200–1800 $m/z$, and loop control was set to "all".

## Computational methods

### Genome annotation and gene set selection

Hg38/GRCh38 was used as a reference genome for all analyses with basic gene annotation based on GENCODE release 40. To define a gene set for the analysis, we selected 10,375 protein-coding genes that satisfied following criteria: (1) did not overlap with another annotated transcript on the same strand and (2) had a 3' end isolated by at least 6 kb from the downstream neighboring gene on the same strand.

### POINT-seq

Quality of POINT-seq reads was checked using FastQC (v0.12.1). Then, raw reads were trimmed using TrimGalore (v0.6.6) in pair-end mode. Reads were mapped to hg38 using STAR (v2.7.6a) (Dobin et al, 2013) with an output of one SAM line for each mapped read, including multimappers (--outSAMmultNmax 1). Output for multiple alignments for each read was set to random (--outMultimapperOrder Random). Aligned reads were assigned to forward and reverse strand using SAMtools (v1.10) (Li et al, 2009). Then, coverage files were created using bedtools (v2.31.1) genomecov tool (Quinlan and Hall, 2010). The coverage files were normalized in R to reads per million (RPMs) and DESeq2 (Love et al, 2014) sizing factors based on ReadsPerGene.out.tab STAR output file. Bigwigs were then generated using bedGraphTo-BigWig tool (Perez et al, 2025).

### T4ph mNET-seq

Quality of mNET-seq reads was checked using FastQC (v0.12.1). Then, raw reads were trimmed using TrimGalore (v0.6.6) in

pair-end. Reads were mapped to hg38 using STAR (v2.7.6a) with an output of one SAM line for each mapped read, including multimappers (--outSAMmultNmax 1). Output for multiple alignments for each read was set to random (--outMultimapper-Order Random). Aligned reads were assigned to forward and reverse strand using SAMtools (v1.10). Then, the custom R script was used to extract the last nucleotide on 3' end of newly transcribed RNA and the coverage files were normalized to reads per million (RPMs) and DESeq2 sizing factors based on ReadsPerGene.out.tab STAR output file. Bigwigs were then generated using bedGraphToBigWig tool.

### POINT-5

POINT-5 was analyzed as previously described in Sousa-Luís et al (2021). In brief, quality of POINT-5 reads was checked using FastQC (v0.12.1). Then, raw reads were trimmed using TrimGalore (v0.6.6) in pair-end. Reads were mapped to hg38 using STAR (v2.7.6a) requiring uniquely mapped reads (–outFilterMultimapNmax 1) and minimum alignment score (–outFilterScoreMin) of 10. Aligned reads were assigned to forward and reverse strand using SAMtools (v1.19.2). Then, the python script developed by Sousa-Luís et al (2021) was used to extract the last nucleotide on 5' end of newly transcribed RNA. These single-nucleotide files were then assigned to forward and reverse strand using SAMtools (v1.19.2) and the coverage files were normalized to reads per million (RPMs) and DESeq2 sizing factors based on ReadsPerGene.out.tab STAR output file. Bigwigs were then generated using bedGraphToBigWig tool.

### Genes classification

To classify genes into two classes, we first generated averaged POINT-seq enrichment profiles for each condition by combining biological replicates with WiggleTools v1.2.11 (wiggletools mean) (Zerbino et al, 2014) and converting the resulting files to bedGraph format. Broad enrichment windows were then identified for each condition using MACS2's bdgbroadcall function (Zhang et al, 2008), applying a minimum signal intensity cutoff to define enriched regions (-c 0.3), a threshold for merging adjacent enriched regions (-C 0.2), and a maximum gap of 3500 base pairs allowed between merged regions (-G 3500). The resulting enrichment windows were used to compare control and SETD2 KO/mut samples.

Active genes were defined based on a minimum POINT-seq signal threshold (RPM > 0.0019) calculated earlier using getCountsByRegions function from BRGenomics v1.12.0 library (Tools for the Efficient Analysis of High-Resolution Genomics Data). Each active gene was extended 5 kb downstream. Enrichment windows overlapping these extended gene coordinates were extracted for both control and SETD2 KO/mut samples. Genes were classified as class II if, after subtracting control enrichment windows from those in the SETD2 KO/mut sample, there remained at least 1 kb of KO/mut-specific enrichment within a region downstream of the annotated gene end. Genes not meeting this criterion were classified as class I. All analyses were performed in R using the rtracklayer v1.54.0 (Lawrence et al, 2009) and plyranges v1.14.0 (Lee et al, 2019) packages compatible with Bioconductor 3.16 release, R v4.2 (BRGenomics library is not supported in the later Bioconductor releases).

### Termination index

The termination index was calculated based on previously averaged POINT-seq files by quantifying the signal (getCountsByRegions

from BRGenomics library) within two regions per gene: an 800 bp window ending on the annotated PAS and a 6 kb window downstream of the annotated PAS. The index was defined as the log10 of ratio of the POINT-seq signal in the 6 kb region to that in the 800 bp region.

### Metagene profiles

Metagene profiles were generated using deepTools v3.5.5 (Ramírez et al, 2016) computeMatrix scale-regions command. Each gene was scaled to a 10,000 bp region, flanked by 5000 bp upstream and 10,000 bp downstream of the annotated TSS and PAS, respectively. Separate bigWig files for forward and reverse strands were provided as input, and the --skipZeros option was applied to exclude regions with zero coverage. For the reverse strand, the --scale -1 option was used to invert the signal, ensuring correct orientation of the data. Matrices for sense and antisense profiles were calculated separately for each strand and then merged using computeMatrixOperations rbind. Metagene plots were generated with the plotProfile command.

### POINT-seq/POINT-5 signal in genic bins

To quantify POINT-seq/POINT-5 signal distribution across genes, each gene was divided into ten consecutive bins of equal width, with bin numbering reflecting gene orientation (i.e., reversed for genes on the minus strand). BAM files were imported as stranded GRanges objects, and for each sample, the number of overlapping reads was counted in each bin using the count_overlaps function from plyranges. The signal in each bin was then normalized to the signal in the first bin of the corresponding gene and replicate. Then, biological replicates were plotted together.

### Cryptic initiation frequency

For the cryptic initiation frequency quantification, we identified genes, which showed increased nascent transcription signal in the gene body in KO vs WT condition, normalized to TSS-proximal signal i.e., relative to level of the gene's normal transcriptional activity. Frequency was estimated using POINT-seq and POINT-5 signals quantified previously across ten equal genic bins. In the first approach, genes were classified as showing increased cryptic activity when the mean signal across bins 2–10 was consistently higher in SETD2 KO than in WT samples in all biological replicates of both POINT-seq and POINT-5. In the second approach, a more stringent filter was applied in which KO > WT was required specifically in the last two genic bins (bins 9 and 10) for every replicate of each dataset. For both strategies, only genes meeting the KO > WT criterion across all corresponding replicates were retained, and the resulting gene sets were intersected between POINT-seq and POINT-5.

### Cleavage index

To calculate the cleavage index, we first identified genes with a single predominant polyadenylation site (PAS). A gene was considered to have one main PAS if at least 90% of the total PAS-associated signal for that gene was concentrated at a single PAS. Using GENCODE v40 annotations, we extracted the coordinates for these genes and extended each region by 500 bp. BAM reads overlapping these extended gene coordinates were then extracted using the SAMtools view command.

For each sample, two BED files were generated based on experimentally determined PAS: one representing a 10 bp "PAS"

window (5 bp upstream and 5 bp downstream of the PAS) and another representing a 250 bp "total" window ending at the PAS. A custom R script was used to sum the BAM reads overlapping these windows, utilizing the count_overlaps_within function from the plyranges package for the PAS window and count_overlaps for the total window. Genes with zero reads in the total region were excluded from further analysis. The cleavage index was then defined as 1 minus the ratio of reads in the PAS window to the total window, or 1 − (nPAS/nTotal).

### 3' mRNA-seq data processing, filtering, and PAS detection

Total RNA was sequenced using the NovaSeq 6000 platform, and raw sequencing data were demultiplexed and converted to FastQ format with bcl2fastq (v2.20.0.422) following Illumina's protocol. Subsequent processing of the FastQ files was carried out according to Lexogen's QuantSeq analysis workflow, including alignment and generation of BAM files. Strand-specific separation of aligned reads was performed using SAMtools (v1.10). Downstream analysis was adapted from the previously published studies (Kamieniarz-Gdula et al, 2019; Derti et al, 2012; Fontes et al, 2017; Rot et al, 2017).

To reduce artifacts from internal priming in A/T-rich genomic regions, a genomic mask was constructed to flag regions containing six or more consecutive adenines (on the forward strand) or thymines (on the reverse strand), as well as any 10-nucleotide window with more than six A's or T's. This mask was refined by unmasking a 20-nucleotide region centered on each gene's annotated 3' end (GENCODE v40) and incorporating experimentally validated polyadenylation sites (PAS) from previous studies to avoid excluding genuine PAS (Derti et al, 2012). Reads overlapping masked regions were filtered out, retaining only those from the most distal nucleotide for downstream analysis. To accurately identify polyadenylation sites (PAS), we utilized the filtered 3' mRNA-seq reads to generate strand-specific, depth-normalized density profiles across the genome. PAS event detection was performed through a systematic, multi-step approach: First, the coverage signals within each 30-nucleotide window were summed, and only those windows with a total signal exceeding 30 were selected for further analysis. Within each qualifying window, the position with the highest signal intensity was identified as the peak, representing a potential PAS. Around each identified peak, a new 30-nucleotide region was defined, and this process was iteratively repeated to resolve and eliminate any overlapping intervals, ensuring that each PAS was uniquely and reliably represented. The resulting set of high-confidence polyadenylation sites was then used to extract PAS usage data for each sample and served as the foundation for downstream analyses of APA usage.

### Alternative polyadenylation (APA) analysis

To assess alternative polyadenylation (APA), differential PAS usage was analyzed using DEXSeq (Anders et al, 2012), incorporating established workflows for APA analysis. All protein-coding genes, as defined in the genome annotation and gene set selection, were included, with gene coordinates extended by 6 kb downstream of annotated 3' ends to enable detection of distal APA events. DEXSeq provided log2 fold change values and adjusted p-values for each PAS, allowing classification of genes with no significant PAS changes (padj ≥ 0.05) as showing no APA shift. For genes with multiple significant PAS (padj < 0.05), the two most differentially used sites were compared; a higher distal-to-proximal usage ratio in

experimental versus control samples indicated a distal shift, while the opposite indicated a proximal shift.

### Mass spectrometry analysis

Raw data files were processed using DIA-NN software, version 2.0.2, to produce a protein group matrix file listing the relative protein abundancies. Prior to this, a spectral library was generated in silico using DIA-NN with a Swiss-Prot reviewed Homo sapiens FASTA file (downloaded from UniProt February 2025) and a FASTA file listing common mass spectrometry contaminants. The additional option "--duplicate-proteins" was used to generate the library. The precursor and protein false discovery rate (FDR) was 1%, default setting. Then, to assess differential protein expression, limma package was used.

### P values and significance test

Statistical test between WT and SETD2 KO/mut condition was conducted using Mann–Whitney test. To calculate significance in overlap between two groups of genes, hypergeometric test was used. P-values: *$p \leq 0.05$, **$p \leq 0.01$, ***$p \leq 0.001$, ****$p \leq 0.0001$.

## Data availability

Data generated in this study is deposited in NCBI GEO under accession numbers GSE301929 and GSE313082. Code used for the analyses is available on https://github.com/STOP-lab/SETD2-in-initiation-and-termination.

The source data of this paper are collected in the following database record: biostudies:S-SCDT-10_1038-S44319-026-00744-1.

## Peer review information

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

## Acknowledgements

Work in K.K.-G. lab was funded by Narodowe Centrum Nauki [2018/30/E/NZ1/00073], European Union - Horizon Europe - European Research Council ERC 'AlternativeEnds' [101042642] and European Molecular Biology Organization EMBO IG 4728-2020. Work in T.N. lab was funded by MEXT/JSPS Kakenhi [JP19K24692, JP24K01957], JST FOREST [JPMJFR2050], Royal Society [ICA\R1\231018], Astellas Foundation for Research on Metabolic

Disorders, Daiichi Sankyo Foundation of Life Science, The Ichiro Kanehara Foundation for Promotion of Medical Sciences and Medical Care, Takeda Science Foundation, The Mitsubishi Foundation, The Mochida Memorial Foundation for Medical and Pharmaceutical Research, The Naito Foundation, The NOVARTIS Foundation (Japan) for the Promotion of Science, Princess Takamatsu Cancer Research Fund, The Shinnihon Foundation of Advanced Medical Treatment Research, The Sumitomo Foundation, and The Uehara Memorial Foundation. CN was supported by JST SPRING [JPMJSP2136], RIKAKEN HD, and JSPS DC2 [JP25KJ1964]. MRG was supported by Polish National Agency for Academic Exchange [PPN/PPO/2019/1/00009]. This work was also supported in part by the MEXT Cooperative Research Project Program, Medical Research Center Initiative for High Depth Omics, and CURE [JPMXP1323015486]. The infrastructure of Omics Science Center Secure Information Analysis System, Medical Institute of Bioregulation at Kyushu University provides the part of computational resource.

## Author contributions

**Magda Kopczyńska**: Formal analysis; Investigation; Visualization; Writing—original draft; Writing—review and editing. **Chihiro Nakayama**: Formal analysis; Investigation; Writing—original draft; Writing—review and editing. **Agata Stępień**: Investigation; Writing—review and editing. **Shoko Ito**: Investigation. **Koshi Imami**: Investigation. **Michał R Gdula**: Formal analysis. **Takayuki Nojima**: Conceptualization; Supervision; Funding acquisition; Writing—review and editing. **Kinga Kamieniarz-Gdula**: Conceptualization; Supervision; Funding acquisition; Writing—original draft; Writing—review and editing.

Source data underlying figure panels in this paper may have individual authorship assigned. Where available, figure panel/source data authorship is listed in the following database record: biostudies:S-SCDT-10_1038-S44319-026-00744-1.

## Disclosure and competing interests statement

The authors declare no competing interests.

# Expanded View Figures

**Figure EV1.    Characteristic features of class I and II genes.**

(**A**) Biological replicates of POINT-seq and T4ph mNET-seq on *VASP* gene in U2OS. (**B**) Principal Component Analysis (PCA) of POINT-seq replicates in U2OS cells. (**C**) Illustration of class II gene definition: POINT-seq enrichment windows were identified and genes were classified as class II when the enrichment window for SETD2 KO or mutation was extended at least 1 kb in comparison to WT. (**D**) PCA for T4ph mNET-seq replicates in U2OS. (**E**) H3K36me3 signal on class I and class II genes in WT condition in U2OS (Wen et al, 2014; data ref: Wen et al, 2014). (**F**) H3K4me3 signal on class I and class II genes (Rane et al, 2024; data ref: Rane et al, 2024). (**G**) H3K27ac signal on class I and class II genes (Wu et al, 2024; data ref: Wu et al, 2024). (**H**) H4K16ac signal on class I and class II genes (Radzisheuskaya et al, 2021; data ref: Radzisheuskaya et al, 2021). (**I**) H3K27me3 signal on class I and class II genes (Jawhar et al, 2025; data ref: Jawhar et al, 2025). (**J**) H3K9me3 signal on class I and class II genes (data ref: Arroyo-Gómez and Reverón-Gómez, 2025). (**K**) Expression comparison between class I and class II genes in U2OS WT based on POINT-seq. (**L**) RNAPII elongation rates measured by 4sU-seq after (4, 8 and 16 min after DRB release), $n$ class I $= 257$, $n$ class II $= 179$ (Baluapuri et al, 2019; data ref: Baluapuri et al, 2019). (**E–L**) All dataset analyzed were derived from U2OS cells. (**M**) Gene length comparison between class I and class II genes in U2OS. (**N**) Intron length comparison (first, inner and last) between class I and class II genes in U2OS. (**O**) Copy number alterations in *SETD2* gene in RCC (blue) and ccRCC (gray). Data from TCGA accessed from https://www.cbioportal.org. (**P**) Expression of *SETD2* in ccRCC in normal (blue) and primary tumor (red), data from TCGA accessed from UALCAN (Chandrashekar et al, 2017). (**Q**) PCA for POINT-seq replicates in RCC. (**R**) POINT-seq replicates on *HK2* gene in RCC. (**S**) Overlap of class II genes between U2OS and RCC models. (**T**) Expression comparison between class I and class II genes in RCC SETD2 WT based on POINT-seq. (**U**) Gene length comparison between class I and class II genes in RCC. (**V**) Intron length comparison (first, inner and last) between class I and class II genes in RCC. Data information: Boxplots on (**K**), (**L**), (**N**), (**P**), (**T**), (**V**) show the median (center line), first and third quartiles (lower and upper bounds of the box), and whiskers extending to the most extreme data points within 1.5×IQR of the lower and upper quartiles, Mann–Whitney test was applied, **$p \leq 0.01$, ****$p \leq 0.0001$. On (**F–K**), (**M**), (**N**) $n$ class I $= 4370$, $n$ class II $= 1399$. On (**T**), (**U**), (**V**) $n$ class I $= 4892$, $n$ class II $= 862$. The results on (**O**) and (**P**) are based upon data generated by the TCGA Research Network: https://www.cancer.gov/tcga.

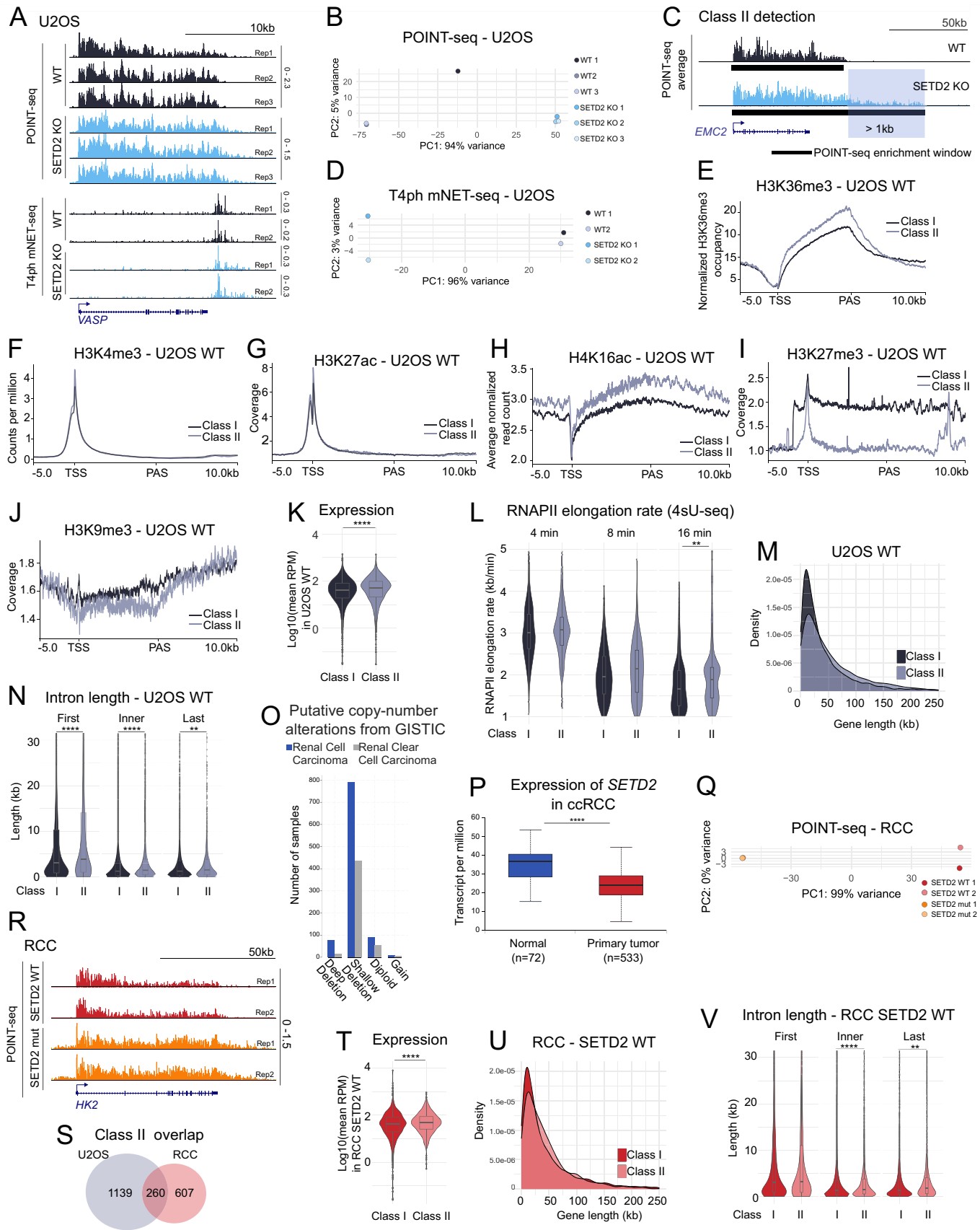

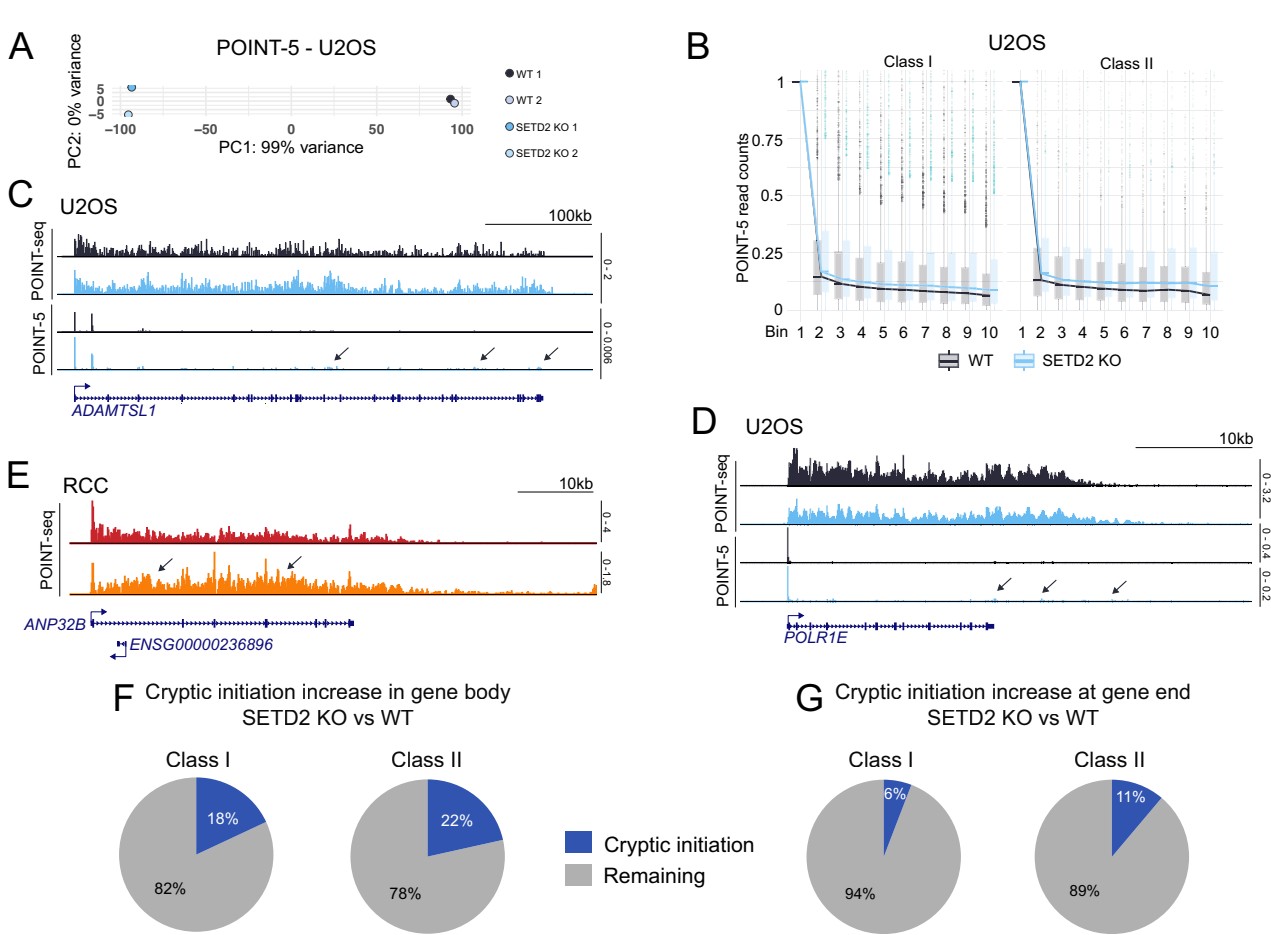

Figure EV2. Cryptic initiation upon SETD2 loss happens more frequently in class II genes.

(A) PCA for POINT-5 biological replicates in U2OS cells. (B) Boxplot representing the number of POINT-5 read counts calculated as described in Fig. 2D for class I, n = 4229 (left) and class II, n = 1359 (right) genes in U2OS WT (black) and SETD2 KO (blue). (C, D) Examples of genes with cryptic initiation in the gene body (indicated by arrows) in U2OS. (E) Example of a gene with cryptic initiation in the gene body (indicated by arrows) in RCC. (F, G) Quantification of genes which show cryptic initiation increase upon SETD2 KO, either in the entire gene body (bins 2–10, F), or specifically at the gene end (bins 9 and 10, G). For each gene, its gene body signal was normalized to TSS-proximal signal (bin 1) to account for changes in normal transcriptional activity following KO (see also Methods). Data information: Boxplot on (B) shows the median (center line), first and third quartiles (lower and upper bounds of the box), and whiskers extending to the most extreme data points within 1.5×IQR of the lower and upper quartiles.

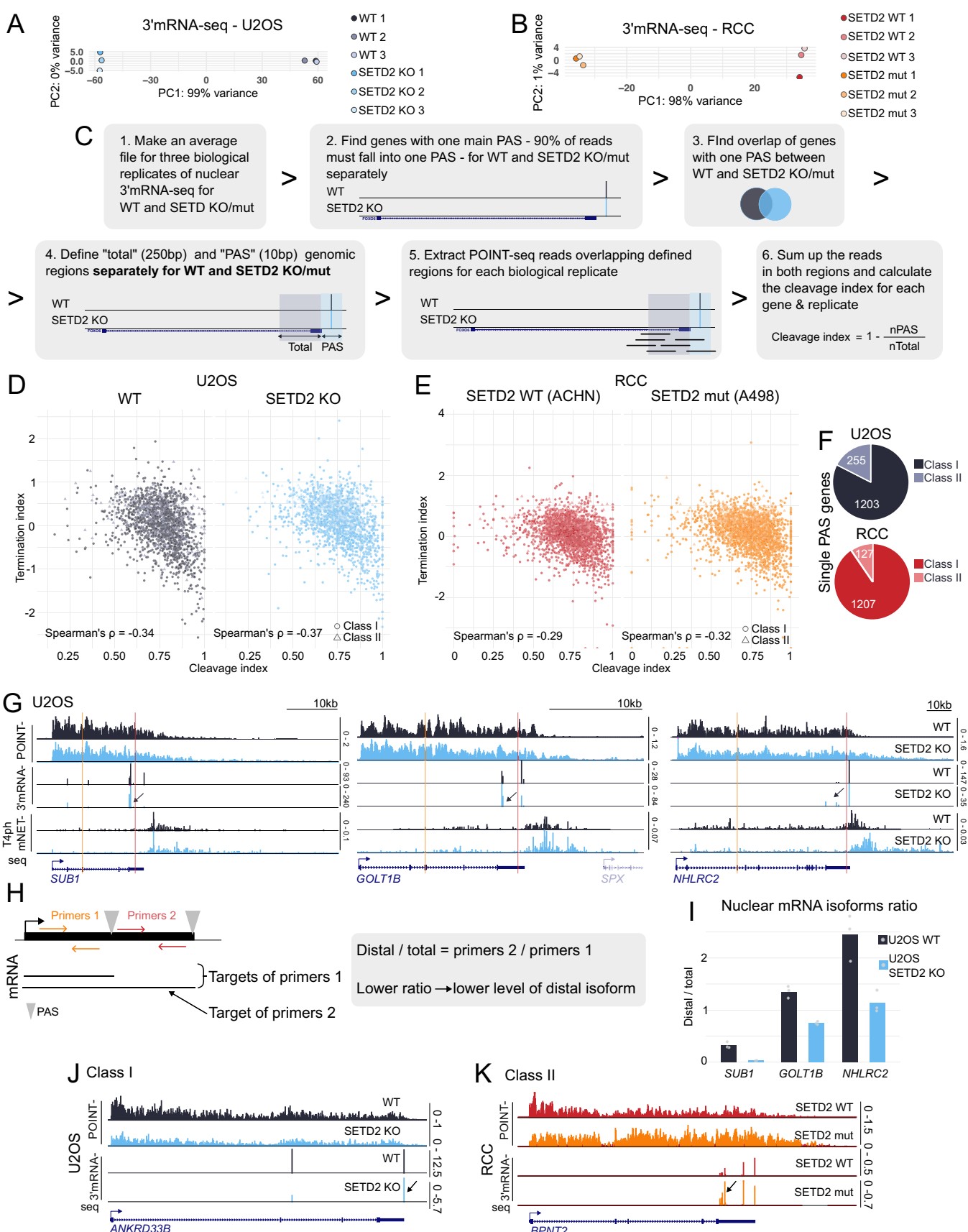

◄ **Figure EV3.  Cleavage index calculation and APA validation.**

(A, B) PCA for 3′mRNA-seq biological replicates in U2OS cells (A) and RCC cells (B) either WT or SETD2 KO/mut. (C) Schematic representation of the cleavage index calculation methodology. (D) Scatterplot representing correlation between termination index and cleavage index in U2OS WT (black) vs SETD2 KO (blue), $n = 1769$. (E) Scatterplot representing correlation between termination index and cleavage index in RCC SETD2 WT (red) vs SETD2 mutation (orange), $n = 2322$. (F) Pie charts of class I and class II gene numbers in single PAS gene category in U2OS (top) and RCC (bottom) models. (G) Genome browser snapshots of genes used for APA validation: *SUB1*, *GOLT1B* and *NHLRC2*. Orange and red highlights represent the regions targeted by the primers. (H) Schematic representation of qPCR assay used for calculating distal/total isoforms ratio. Primers were designed to target all isoforms (primers 1) and the most distal isoform (primers 2). Distal to total ratio was calculated by dividing the amount of product targeted by primers 2 by product of primers 1. (I) Distal/total ratio of nuclear mRNA isoforms for *SUB1*, *GOLT1B* and *NHLRC2* genes. Each biological replicate is represented by a dot ($n = 3$). (J) Gene example of class I gene with distal APA (indicated by an arrow) in U2OS. (K) Gene example of class II gene with proximal APA (indicated by an arrow) in RCC.

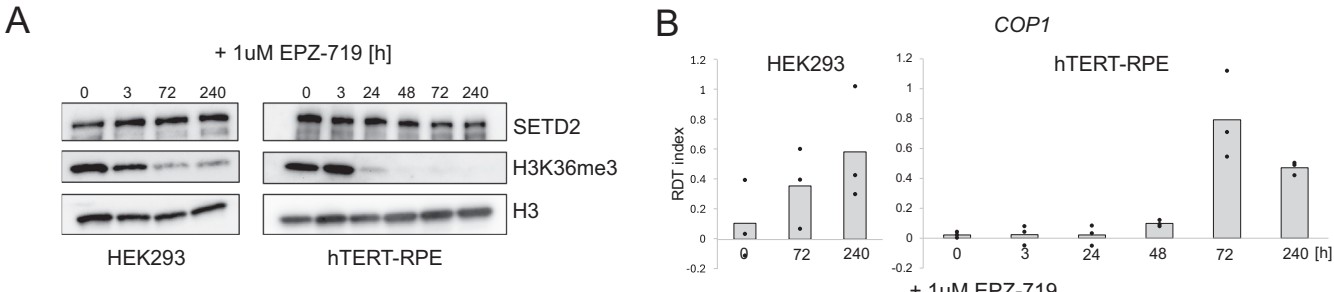

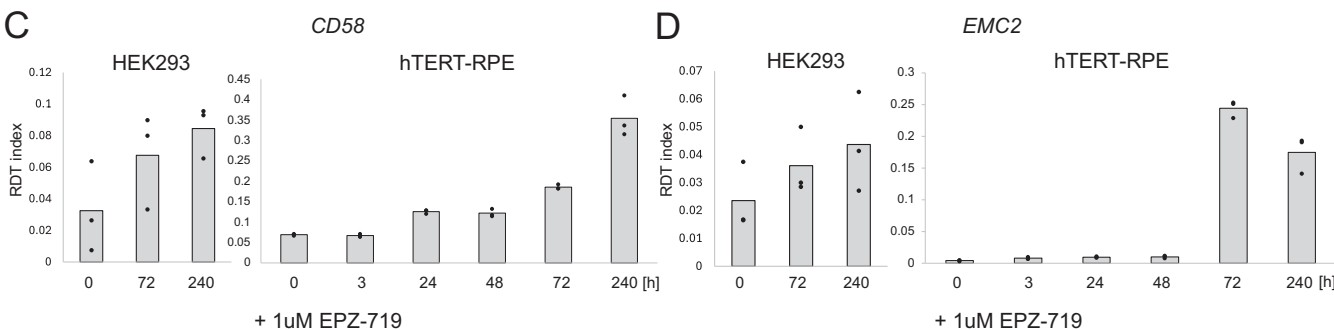

Figure EV4. SETD2 inhibition leads to transcription readthrough induction also in non-cancerous cells.

(A) Western Blot of SETD2, H3K36me3 and H3 in HEK293 (left) and hTERT-RPE cells (right) treated with the SETD2 inhibitor EPZ-719 for the indicated number of hours.
(B–D) RDT index calculated for *COP1*, *CD58* and *EMC2* genes in HEK293 (left) and hTERT-RPE (right) cells treated with EPZ-719 for the indicated number of hours.
Individual dots represent distinct biological replicates ($n = 3$). Source data are available online for this figure.

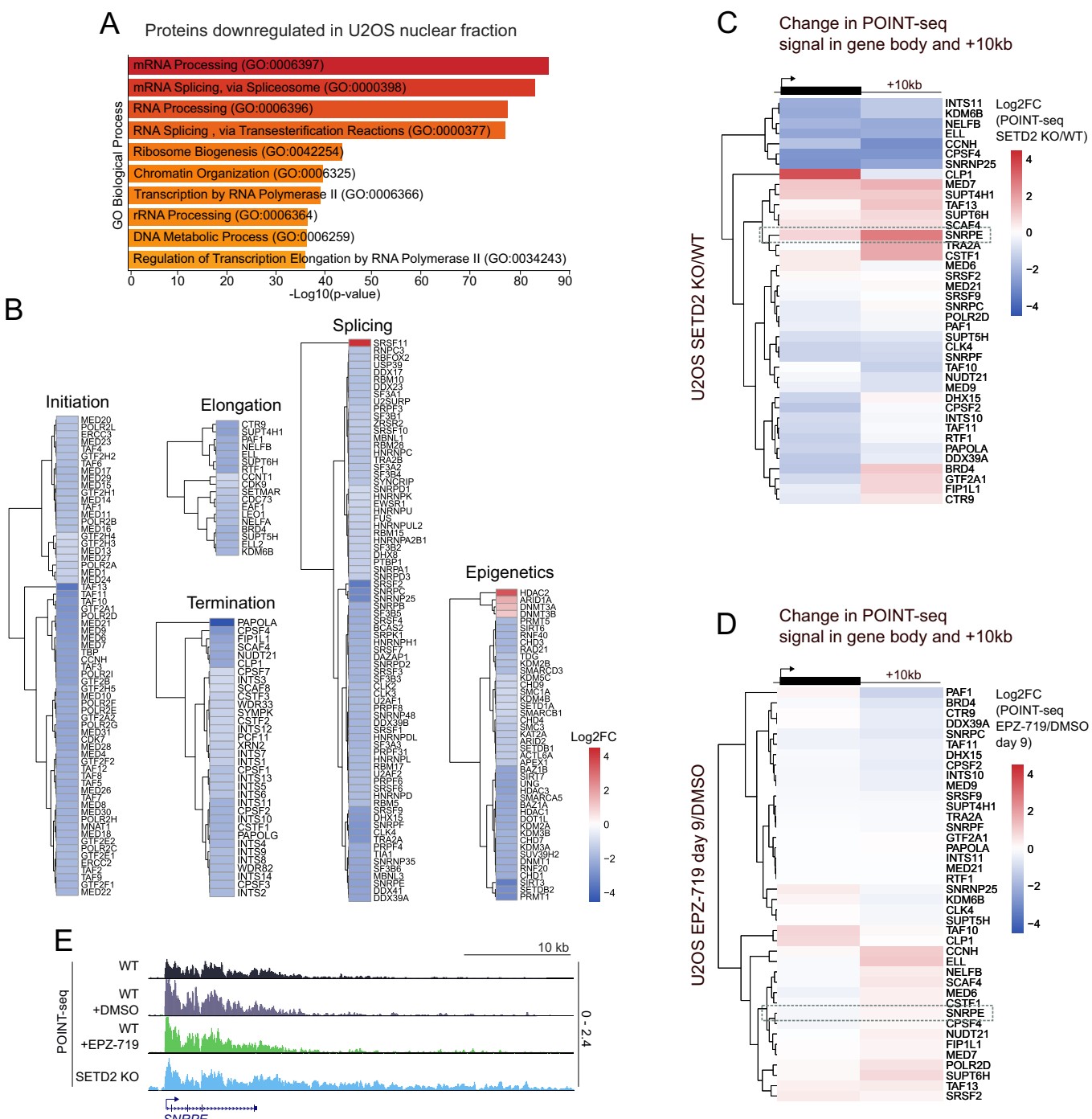

**Figure EV5. SETD2 loss leads to downregulation of key transcription-related proteins in the nuclear fraction.**

(A) Gene ontology results representing biological process that proteins downregulated in U2OS nuclear fraction are involved in. (B) Heatmaps representing log2fc of protein abundance in SETD2 KO vs WT in nuclear fraction of U2OS cells. Proteins were divided into groups corresponding to their involvement in different stages or transcriptional control. (C) Heatmap representing change in POINT-seq signal in gene body and 10 kb downstream of the annotated gene end in U2OS SETD2 KO/WT. (D) Heatmap representing change in POINT-seq signal in gene body and 10 kb downstream of the annotated gene end in U2OS WT + DMSO/WT + EPZ-710. (E) Gene example of POINT-seq signal (marked on (C) and (D) by a dotted rectangle) in U2OS WT (black), WT + DMSO (gray), WT + EPZ-719 (green) and SETD2 KO (blue). Data information: for (A) Fisher's exact test was applied.

