## [Peer Review File · EMBO Reports]

SETD2 methyltransferase activity promotes correct transcription initiation and termination

Magda Kopczyńska, Chihiro Nakayama, Agata Stepień, Shoko Ito, Koshi Imami, Michał Gdula, Takayuki Nojima, and Kinga Kamieniarz-Gdula

Corresponding author(s): Kinga Kamieniarz-Gdula (kinga.kamieniarz-gdula@amu.edu.pl) , Takayuki Nojima (taka.nojima@bioreg.kyushu-u.ac.jp)

Review Timeline:

Submission Date:	18th Jul 25
Editorial Decision:	15th Sep 25
Revision Received:	16th Dec 25
Editorial Decision:	29th Jan 26
Revision Received:	6th Feb 26
Accepted:	27th Feb 26

Editor: Esther Schnapp

Transaction Report:

Dear Dr. Kamieniarz-Gdula,

Thank you for the submission of your manuscript to EMBO reports. We have now received the full set of referee reports that is pasted below.

As you will see, the referees acknowledge that the findings are potentially interesting. However, they also raise some concerns and have several suggestions for how the study could be improved and strengthened. I think all suggestions are good and should be addressed. I also agree with referee 2 that the role of SETD2 should be examined in wildtype cells as well, and referee 3 hints at this as well. Please let me know if you have any comments or questions and we can discuss the exact revision requirements further, also in a video chat, if you like.

I would thus like to invite you to revise your manuscript with the understanding that the referee concerns must be fully addressed and their suggestions taken on board. Please address all referee concerns in a complete point-by-point response. Acceptance of the manuscript will depend on a positive outcome of a second round of review. It is EMBO reports policy to allow a single round of major revision only and acceptance or rejection of the manuscript will therefore depend on the completeness of your responses included in the next, final version of the manuscript.

We realize that it is difficult to revise to a specific deadline. In the interest of protecting the conceptual advance provided by the work, we recommend a revision within 3 months (16th Dec 2025). Please discuss the revision progress ahead of this time with the editor if you require more time to complete the revisions.

- 1) A data availability section providing access to data deposited in public databases is missing. If you have not deposited any data, please add a sentence to the data availability section that explains that.
- 2) Your manuscript contains statistics and error bars based on $n=2$. Please use scatter blots in these cases. No statistics should be calculated if $n=2$.

3) We replaced Supplementary Information with Expanded View (EV) Figures and Tables that are collapsible/expandable online. A maximum of 5 EV Figures can be typeset. EV Figures should be cited as 'Figure EV1, Figure EV2' etc... in the text and their respective legends should be included in the main text after the legends of regular figures.

5) a complete author checklist, which you can download from our author guidelines . Please insert information in the checklist that is also reflected in the manuscript. The completed author checklist will also be part of the RPF.

6) Please note that all corresponding authors are required to supply an ORCID ID for their name upon submission of a revised manuscript (). Please find instructions on how to link your ORCID ID to your account in our manuscript tracking system in our Author guidelines

7) Before submitting your revision, primary datasets produced in this study need to be deposited in an appropriate public database (see <https://www.embopress.org/page/journal/14693178/authorguide#datadeposition>). Please remember to provide a reviewer password if the datasets are not yet public. The accession numbers and database should be listed in a formal "Data Availability" section placed after Materials & Method (see also <https://www.embopress.org/page/journal/14693178/authorguide#datadeposition>). Please note that the Data Availability Section is restricted to new primary data that are part of this study. * Note - All links should resolve to a page where the data can be accessed. *
If your study has not produced novel datasets, please mention this fact in the Data Availability Section.

- the name of the statistical test used to generate error bars and P values,
- the number (n) of independent experiments (please specify technical or biological replicates) underlying each data point,
- the nature of the bars and error bars (s.d., s.e.m.),
- If the data are obtained from n {less than or equal to} 2, use scatter blots showing the individual data points.

12) All Materials and Methods need to be described in the main text using our 'Structured Methods' format, which is required for all research articles. According to this format, the Methods section includes a separate Reagents and Tools Table file (listing key reagents, experimental models, software and relevant equipment and including their sources and relevant identifiers) and a Methods and Protocols section describing the methods using a step-by-step protocol format. The aim is to facilitate adoption of the methodologies across labs. More information on how to adhere to this format as well as a downloadable template (.docx) for the Reagents and Tools Table can be found in our author guidelines:
<https://www.embopress.org/page/journal/14693178/authorguide#structuredmethods>.

An example of a Method paper with Structured Methods can be found here: <https://www.embopress.org/doi/full/10.1038/s44320-024-00037-6#sec-4>

As part of the EMBO publication's Transparent Editorial Process, EMBO reports publishes online a Review Process File (RPF)

to accompany accepted manuscripts. This File will be published in conjunction with your paper and will include the referee reports, your point-by-point response and all pertinent correspondence relating to the manuscript.

I look forward to seeing a revised form of your manuscript when it is ready.

Kind regards,
Esther

Referee #1:

The study by Nakayama et al., investigates role of SETD2 and its methyltransferase activity in transcription by employing a range of the state of the art transcriptomic approaches (mNET seq, POINT seq, 3' RNA seq) to assess consequences of either loss of SET2D protein or its activity in previously generated SET2D KO and SET2D catalytic mutant U2OS and RCC cell lines. The authors delineate functional contribution of SET2D to transcription proposing that there are at least two main classes of genes that are differentially affected by loss of SET2D -Class I and II. They propose that Class I depends on SET2D for transcription initiation whereas Class II relies on SET2D for timely transcription termination in both cell lines. Consistently with the previous reports by Fred Winston and others there is also cryptic transcription initiation observed within the gene body of selected genes.

To gain further mechanistic insight into defective termination authors map cleavage sites using 3'end RNA seq and termination sites using ChIP-seq for Thr4-P Pol II. They assess efficiency of pre-mRNA 3'end cleavage and propose that transcription read-through at class II genes is due to defective cleavage. Surprisingly, 3' RNA seq analyses revealed potential use of the proximal PAS associated with the 3' Read-through transcription and extended termination zone (Thr4-P distribution) observed suggesting that APA may not contribute to failure to terminate transcription in the absence of functional SET2D.

Finally, using H3K36me inhibitor (EPZ-719) doesn't fully recapitulate effects observed upon SET2D loss suggesting that SET2D is likely to contribute to transcription termination indirectly possibly via altered abundance of the transcription machinery components observed in SET2D KO.

Overall, it is a nice and systematic study that advance our understanding of contribution of SET2D and H3K36 methylation to RNA pol II transcription. The study would further benefit from some additional experiments /analyses in order to address some of the points outlined below.

1. Does the cleavage efficiency analyses in Fig 3 D and E takes into an account APA? Reduction in cleavage efficiency at the canonical PAS may be indicative that the alternative PAS is used.
2. I am not fully convinced that it is appropriate to refer to the regions with high Thr4-P as 'termination zone'. Rather, it should be viewed as region downstream of Cleavage polyadenylation machinery recruitment associated with Ser5 de-phosphorylation by Ssu72 phosphatase that masks Thr4-P from detection within the gene body region as was described in Moreno et al (doi: 10.1126/sciadv.adq0350).
3. It is somewhat surprising that Thr4-P zone (Fig 4B) is not shifted closer to the proximal PAS in SET2D KO. It would be good to use an alternative approach to validate 3' RNA seq data at least on a couple of representative genes?
4. What is the frequency of cryptic internal initiation in SETD2 KO?
5. Perhaps it could be worth to include samples treated for a shorter time with EPZ-719? Are other transcriptional defects seen in SETD2 recapitulated upon exposure to EPZ-719?

Referee #2:

The study by Nakayama et al. investigates the role of the human histone H3K36 methyltransferase Setd2 in transcriptional

elongation and genome stability using POINT-seq and T4ph-mNET-seq techniques. They identify two distinct classes of genes that respond differently to Setd2 depletion: Class I genes exhibit initiation defects, while Class II genes face termination defects. The focus on Class II genes demonstrates how Setd2's enzymatic activity affects their transcription readthrough and 3'-end processing.

While the study provides valuable genome-wide insights into Setd2's differential impact on transcription, the analysis could benefit from a deeper exploration of the specific characteristics and functional implications of Class I versus Class II genes. This additional detail would enhance the biological significance of the results. Moreover, the use of three tumor cell lines instead of wild-type cells raises concerns about the generalizability of the findings, particularly since the response to Setd2 depletion might vary between osteosarcoma and RCC cells. Addressing these divergent responses would clarify the study's applicability. Furthermore, establishing the clinical relevance of the affected gene classes could significantly amplify the impact of the research.

Comments:

1. The method used by the authors to analyze POINT-seq data may not adequately identify cryptic transcription events. Authors should use techniques such as northern blotting or more precise 5'- and 3'-end RNA mapping methods to confirm the presence of cryptic initiation phenotypes.
2. Given that Setd2 deletion is known to impact longer genes more significantly, an analysis detailing the characteristics (such as length, expression levels, and chromatin states) of Class I and Class II genes could shed light on the underlying mechanisms affected by Setd2.
3. The small overlap between genes affected by Setd2 deletion and those affected by catalytic mutation, as shown in Figure EV1M, raises questions about the robustness of the findings. Including data from Setd2-depleted normal cells, in addition to tumor cells, could provide a clearer understanding of Setd2's role.
4. Figure 2E/F highlights significant differences in cryptic transcription between U2OS and RCC cells, indicating different mechanisms may be at play in Setd2 deletion versus catalytic mutation. The authors should explore and discuss potential reasons for these differences, such as the possibility of dominant-negative effects from the mutant.
5. While Setd2 is well-documented for its role in Pol II elongation, the authors should propose a model explaining how it might also influence transcription initiation. Additionally, experiments to demonstrate that initiation defects could be indirectly caused by known issues with Setd2 mutants, such as cryptic initiation or antisense transcription, would strengthen this hypothesis.
6. Investigating whether genes involved in osteosarcoma or ccRCC tumorigenesis are enriched in either Class I or Class II could provide insights into Setd2's disease-specific roles, enhancing our understanding of its function in cancer development.

Referee #3:

SETD2 deposits H3K36me3 in the bodies of active genes and is positively correlated with transcription. Importantly, while the effects of SETD2 on transcription have been previously investigated, this did not include nascent transcription analysis as well as a direct assessment of engaged RNA PolII dynamics. Nakayama et al. address this gap in detail through a series of sophisticated techniques for measurement of RNA PolII-engaged RNA applied to mutant cancer cell lines.

The main conclusion is that SETD2 regulates both transcription initiation and termination, where it further can regulate cleavage as well as read-through. Most of the experimental evidence is based on two cancer types, namely i) a U2OS cell line in which SETD2 has been deleted and ii) two patient-derived RCC cell lines with and without a catalytic mutation of SETD2. While stable mutants and variable genetic backgrounds do not allow for conclusions regarding the direct effects of SETD2 loss, the authors acknowledge this limitation and additionally use a catalytic inhibitor to test their findings. This experiment corroborates the overall conclusions on the one hand, but also provides evidence for widespread indirect effects.

The addressed question is important and the authors' approach provides many interesting insights using state-of-the-art experimental methods, data and analysis with the highest quality. Similarly, the manuscript is very-well written, clear and concise, with clear figures that are appropriately interpreted. While the manuscript's core strength lies in its thorough analysis and technical approach, as well as in the use of a combination of methods that actually allow to clarify the transcriptional effects, mechanistic conclusions cannot be fully drawn from the here chosen experimental system. My major concern is that the usage of stable mutant cell lines with variable parental background may interfere with causal conclusions and the direct effects of H3K36-methyltransferase inhibition remain unclear. I believe the points below will strengthen the main message of the manuscript.

Major comments:

1. Usage of cancer cell lines as described here may be problematic for two reasons:
 - a) RCC cell lines are derived from two different patients and therefore likely are both genetically and cell-type-wise not directly

comparable, yet treated here as wildtype versus knockout. This makes the magnitude of the effects obtained in the RCC cell lines incomparable with the effects in the U2OS cell lines that have the same genetic background. Going forward, I recommend to the authors to downplay this comparison and acknowledge the limitation.

b) The knockout/mutation are stable, therefore the effects are likely to be indirect. To strengthen the main message, it would be important to investigate direct effects beyond the effects reported in Fig. 4 (see below).

2. Direct effects are tested by using a methyltransferase inhibitor in Fig. 4, whereby unsurprisingly the effects achieved at tested genes are smaller than in the stable cell lines. Through investigating the proteome, the authors then conclude that the majority of SETD2-mediated effects on transcription are indirect, since the inhibitor affects RNA PolIII and other RNA processing enzymes at the protein level. To contextualize these effects and discuss the potential indirect effects further, the authors should address the following two points:

a) How does SETD2 inhibition affect the proteins responsible for RNA processing? Is there a direct effect upon methyltransferase activity inhibition on the readthrough/termination of the genes encoding proteins involved in RNA processing/RNA PolIII?

b) Measure H3K36me3 before/after treatment to test if the observed effects on cleavage/readthrough correlate with alterations in the methyltransferase activity and are therefore likely to be direct.

Minor comments:

3. Were the genome-wide experiments calibrated (using spike-ins)? If not, this could distort potential global effects.

4. Typo on p.9, paragraph heading: 'The effects of SETD2 activity on transcription readthrough is...' should be 'are

Reviewer responseGeneral response:

We thank the reviewers for their constructive, insightful and helpful suggestions, which helped to substantially improve the original manuscript.

We were also pleased to read their positive comments: "Overall, it is a nice and systematic study that advance our understanding of contribution of SET2D and H3K36 methylation to RNA pol II transcription (Referee 1)"; "the study provides valuable genome-wide insights into Setd2's differential impact on transcription" (Referee 2); and "The addressed question is important and the authors' approach provides many interesting insights using state-of-the-art experimental methods, data and analysis with the highest quality" (Referee 3).

In our attached revised manuscript, we have accommodated all the reviewers' suggestions.

To assess the robustness of SETD2 methyltransferase activity on transcription termination, we treated two non-cancerous cell lines (HEK293 and hTERT-RPE) with the SETD2 inhibitor EPZ-719 and measured readthrough induction in several representative genes (Fig EV4). To examine the global impact of SETD2 inhibition on transcription, we also performed POINT-seq on our main experimental model, U2OS cells, after 9 days of EPZ-719 treatment (Fig 4).

Because we found that readthrough is independent of alternative polyadenylation, we validated our 3' mRNA-seq results by qPCR on a set of representative genes (Fig EV3G-I). We further compared class I and class II genes in more detail, including an analysis of several histone marks associated with active and inactive transcriptional states (Fig EV1E-J). In addition, we examined cryptic transcription initiation and estimated its frequency in both gene classes (Fig EV2F-G).

Finally, we investigated transcriptional changes in genes encoding proteins involved in transcription and RNA processing that are downregulated upon SETD2 loss, analyzing both SETD2 KO and SETD2 inhibition conditions (Fig EV5C-E).

We believe the revised manuscript will be of value to the EMBO Reports community.

Our point-to-point response to the reviewers' comments can be found below.

Referee #1:

The study by Nakayama et al., investigates role of SETD2 and its methyltransferase activity in transcription by employing a range of the state of the art transcriptomic approaches (mNET seq, POINT seq, 3' RNA seq) to assess consequences of either loss of SET2D protein or its activity in previously generated SET2D KO and SET2D catalytic mutant U2OS and RCC cell lines. The authors delineate functional contribution of SET2D to transcription proposing that there are at least two main classes of genes that are differentially affected by loss of SET2D -Class I and II. They propose that Class I depends on SET2D for transcription initiation whereas Class II relies on SET2D for timely transcription termination in both cell lines. Consistently with the previous reports by Fred Winston and others there is also cryptic transcription initiation observed within the gene body of selected genes.

To gain further mechanistic insight into defective termination authors map cleavage sites using 3'end RNA seq and termination sites using ChIP-seq for Thr4-P Pol II. They assess efficiency of pre-mRNA 3'end cleavage and propose that transcription read-through at class II genes is due to defective cleavage. Surprisingly, 3' RNA seq analyses revealed potential use of the proximal PAS associated with the 3' Read-through transcription and extended termination zone (Thr4-P distribution) observed suggesting that APA may not contribute to failure to terminate transcription in the absence of functional SET2D.

Finally, using H3K36me inhibitor (EPZ-719) doesn't fully recapitulate effects observed upon SET2D loss

suggesting that SET2D is likely to contribute to transcription termination indirectly possibly via altered abundance of the transcription machinery components observed in SET2D KO.

Overall, it is a nice and systematic study that advance our understanding of contribution of SET2D and H3K36 methylation to RNA pol II transcription. The study would further benefit from some additional experiments /analyses in order to address some of the points outlined below.

Thank you for the kind words!

1. Does the cleavage efficiency analyses in Fig 3 D and E takes into an account APA? Reduction in cleavage efficiency at the canonical PAS may be indicative that the alternative PAS is used.

We agree that this is an important point. To make sure we look at changes in cleavage and not in alternative polyadenylation, for this analysis we have chosen genes where mainly one PAS is used. To anticipate the situation in which main PAS is shifted between the conditions, we defined "PAS windows" separately for each condition. In order to make this clear for the readers, we have now added a schematic of our algorithm in Fig EV3C.

2. I am not fully convinced that it is appropriate to refer to the regions with high Thr4-P as 'termination zone'. Rather, it should be viewed as region downstream of Cleavage polyadenylation machinery recruitment associated with Ser5 de-phosphorylation by Ssu72 phosphatase that masks Thr4-P from detection within the gene body region as was described in Moreno et al (doi: 10.1126/sciadv.adq0350).

The evidence and arguments for usage of regions with T4ph-modified polymerase as termination zones has been provided by us in our recent paper: <https://doi.org/10.1093/nar/gkae1240> The paper by Moreno et al. doi: 10.1126/sciadv.adq0350 suggests that T4ph levels at the TSS are hugely underestimated due to S5ph occlusion. We do not find the data on occlusion convincing as (i) the antibody 6D7 used by the authors for western blot is non-linear in this application, so the quantifications not reliable; (ii) Ssu72 phosphatase specificity was not tested in a cell lysate and the distribution of other phospho-marks upon Ssu72 treatment has not been checked; (iii) a cytoplasmic protein was used as loading control and western blot normalization factor in nuclear extracts; and (iv) 20-fold more single-phosphorylated rather than double-phosphorylated T4ph peptides were found in mass spectrometry (DOI: 10.1016/j.molcel.2015.12.003). In our view, occlusion is unlikely to occur in vivo, since single heptad repeats typically carry only one phospho-mark in vivo; and in light of the above issues, an antibody should be generated to verify the existence, abundance and function of a putative T4phS5ph double-phosphorylated Pol II bound to the TSS, which for now is insufficiently supported by experimental evidence.

We hope this explanation convinces the referee.

Otherwise, if our arguments are not accepted by the referee, we will discuss the possibility of T4ph masking, citing Moreno et al. in our paper.

3. It is somewhat surprising that Thr4-P zone (Fig 4B) is not shifted closer to the proximal PAS in SET2D KO. It would be good to use an alternative approach to validate 3' RNA seq data at least on a couple of representative genes?

We agree with the reviewer that the APA result is somewhat surprising (and to us exciting). As requested, we validated our genome-wide approach using qPCR on 3 representative genes. For each gene, we designed two primer pairs: first targets all isoforms of the transcript, second only the most distal one (Fig EV3G-H). Then, we calculated the distal/total ratio for both U2OS WT and SETD2 KO –

lower ratio indicates lower abundance of the most distal isoform. The results mirror and confirm our 3' mRNA-seq results and are now shown in Fig EV3I.

Related to our finding, in November a preprint came out from Neugebauer lab that shows readthrough transcription in the absence of changes in RNA processing (which occur with delay) in an entirely different system (treatment of CML cells by imatinib) using an independent approach: PacBio long read sequencing <https://pubmed.ncbi.nlm.nih.gov/41279586/>.

4. What is the frequency of cryptic internal initiation in SETD2 KO?

Roughly 20% (18% for class I genes and 21.5% for class II genes).

For the quantification, we identified genes, which showed increased nascent transcription signal in the gene body in KO vs WT condition, normalized to TSS-proximal signal i.e. relative to level of the gene's normal transcriptional activity. For stringency, the increase had to be present in both POINT-seq and in POINT-5, in all biological replicates.

We also repeated the analysis, considering only cryptic initiation in the last two bins (normalized to TSS-linked initiation), reasoning that aberrant initiation closer to gene ends is more likely to contribute to readthrough transcription. This is 5.8% for class I genes, and 11.15% for class II genes.

This new analysis is now included in Fig EV2F-G.

5. Perhaps it could be worth to include samples treated for a shorter time with EPZ-719?

We did not observe any effects on U2OS cells with treatment with EPZ-719 of 3 days despite the loss of H3K36me3(Fig 4A-E). However, we have now also performed SETD2 inhibition on non-cancerous cells and found that treatment of hTERT-RPE cells with EPZ-719 gives faster readthrough effects (Fig EV4), at least for three tested genes.

This analysis is now included in Fig EV4.

Are other transcriptional defects seen in SETD2 recapitulated upon exposure to EPZ-719?

To answer this question, we now additionally performed POINT-seq on the main cellular model in our manuscript, U2OS cells, treated with EPZ-719 for 9 days (i.e. at the time point when the readthrough phenotype could be detected in qPCR), in addition to SETD2 KO. Similarly to our previously reported results using qPCR, our new sequencing data confirmed that a readthrough effect after 9 days of SETD2 inhibition is detectable, but much milder than after KO (new Fig 4F). The strongest other transcriptional defect upon SETD2 KO was defective initiation, especially in class I genes (Fig 1C). Notably, EPZ-719 treatment for 9 days did not decrease initiation, neither of class I nor class II gene. The new data suggest that SETD2 depletion induced read-through in class II genes occurs upstream of the initiation defect.

Those new experiments and analysis are now included in Fig 4.

Referee #2:

The study by Nakayama et al. investigates the role of the human histone H3K36 methyltransferase Setd2 in transcriptional elongation and genome stability using POINT-seq and T4ph-mNET-seq techniques. They identify two distinct classes of genes that respond differently to Setd2 depletion: Class I genes exhibit initiation defects, while Class II genes face termination defects. The focus on Class II genes demonstrates how Setd2's enzymatic activity affects their transcription readthrough and 3'-end processing.

While the study provides valuable genome-wide insights into Setd2's differential impact on

transcription, the analysis could benefit from a deeper exploration of the specific characteristics and functional implications of Class I versus Class II genes. This additional detail would enhance the biological significance of the results. Moreover, the use of three tumor cell lines instead of wild-type cells raises concerns about the generalizability of the findings, particularly since the response to Setd2 depletion might vary between osteosarcoma and RCC cells. Addressing these divergent responses would clarify the study's applicability. Furthermore, establishing the clinical relevance of the affected gene classes could significantly amplify the impact of the research.

Comments:

1. The method used by the authors to analyze POINT-seq data may not adequately identify cryptic transcription events. Authors should use techniques such as northern blotting or more precise 5'- and 3'-end RNA mapping methods to confirm the presence of cryptic initiation phenotypes.

We agree that POINT-seq is not ideally suited for detecting cryptic initiation events, even if superior to conventional RNA-seq which was used for this purpose before (Carvalho et al., NAR 2013 doi: 10.1093/nar/gks1472).

Therefore, we also performed experiments using combination of the POINT-5 technique and 5' phosphate-dependent RNA exonuclease, which is as specific as CAGE, and even more sensitive in detecting 5' ends of unstable, transient transcripts such as PROMPTs.

Below we attach a figure from (Souza-Luis et al. Mol Cell 2021, doi: 10.1016/j.molcel.2021.02.034) displaying the comparison of POINT-5 and CAGE.

Figure 1 from (Souza-Luis et al. Mol Cell 2021) removed

(E) Heatmaps of POINT-5 and CAGE (untreated or RRP40KD) signals for PROMPT-mRNA pairs at TSS. KD, knockdown or depletion. Scaled transcripts per million (TPM) is shown for sense (+) (blue) and antisense (-) (red).

RRP40 is a component of RNA exosome complex which degrades lncRNAs such as a PROMPT.

(F) Quantitation of (E).

(G) Heatmaps of POINT-5 and CAGE (untreated or RRP40KD) signals for mRNA-mRNA pairs at TSS. Scaled TPM is shown for sense (+) (blue) and antisense (-) (red).

(H) Quantitation of (G)."

To strengthen this important methodological point, we put more emphasis in the manuscript on POINT-5, mentioning it is as sensitive as CAGE, and performed additional quantification of cryptic initiation events using both POINT-seq and POINT-5 data.

2. Given that Setd2 deletion is known to impact longer genes more significantly, an analysis detailing the characteristics (such as length, expression levels, and chromatin states) of Class I and Class II genes could shed light on the underlying mechanisms affected by Setd2.

Thank you for this suggestion. While we had previously looked at some features such as gene length in Fig EV1, we didn't present any analysis regarding chromatin modifications. Our extended analysis now shows that class II genes compared to class I are indeed longer (Fig EV1M,U), more highly expressed (Fig EV1K,T), carry more active and less inactive marks (Fig EV1E-J), and have a faster elongation rate as measured by 4sU-seq (Balupuri et al, 2019, Fig EV1L). The above features have been determined for U2OS cells, which are more researched and more data is available. For RCC cells, less genome-wide data is available, but analysis of the gene characteristics where it could be tested (gene length and activity) shows the same trend.

Those data are now presented in Fig EV1F-L.

3. The small overlap between genes affected by Setd2 deletion and those affected by catalytic mutation, as shown in Figure EV1M, raises questions about the robustness of the findings. Including data from Setd2-depleted normal cells, in addition to tumor cells, could provide a clearer understanding of Setd2's role.

From our point of view, the observed overlap is expected. A gene can belong to class I in one type of cell, and to class II in a different cell type. Whether a gene belongs to class I or II is defined by us by the presence or absence of readthrough – this differs in the cell lines. As mentioned in response to above comment 2, class II genes are associated with higher transcriptional activity, elongation speed and more active chromatin environment. Osteosarcoma cells and RCC cells have different expression profiles of the same genes and different chromatin landscape - even in cells with wild-type SETD2. Further on, in osteosarcoma cells we analyze a lab-generated KO, and for RCC a catalytic mutation occurring naturally in a patient.

To make this clearer to the readers we now include the following sentence in the manuscript: "We interpret the limited overlap between the models to be a result of osteosarcoma and RCC cells having different transcriptional profiles, and class II genes being associated with higher expression levels." (page 7 in the revised manuscript)

Further on, following the referee's suggestion and to make our study more robust, we corroborated the role of SETD2 in readthrough transcription also in normal cells. We obtained non-cancerous cell lines (hTERT-RPE and HEK293 cells) and treated them with EPZ-719 to investigate whether we can see the same termination delay as we did in U2OS, cancer-derived cells. First, to evaluate EPZ-719 inhibition on SETD2 methyltransferase activity on these cell lines, we performed western blotting (Fig EV4A). The Western blot analysis of H3K36me3 showed no significant change after 3 hours of SETD2 inhibition; however, after 24 hours, the signal had almost completely disappeared. At all subsequent time points, the H3K36me3 signal remained nearly undetectable. Initially, RT-qPCR was performed in hTERT-RPE and HEK293 cells at the same time points (3 days and 9 days) as those we used for U2OS cells. For the analysis, we chose genes which belong to class II in osteosarcoma and are active also in hTERT-RPE and HEK293 cells. In HEK293 cells, the RDT index showed a slight increase, but the change was not as pronounced as observed in U2OS cells. In contrast, and rather unexpectedly, hTERT-RPE cells exhibited a significant increase in the RDT index already after 3 days of inhibition. Therefore, shorter time points (3 h, 24 h, and 48 h) were additionally examined. The analysis revealed that at these earlier time points, the RDT index showed only a marginal or no increase. These results indicate that the correlation between H3K36me3 depletion and RDT varies among cell types. Furthermore, even in cases where such a correlation exists, the RDT increase does not occur immediately after H3K36me3 depletion. This temporal time lag suggests that H3K36me3 loss is not directly responsible for the observed Pol II termination delay.

Those new experiments are now included in Fig EV4.

4. Figure 2E/F highlights significant differences in cryptic transcription between U2OS and RCC cells, indicating different mechanisms may be at play in Setd2 deletion versus catalytic mutation. The authors should explore and discuss potential reasons for these differences, such as the possibility of dominant-negative effects from the mutant.

Our use of the RCC cell lines to represent catalytically inactive SETD2 inevitably introduces discrepancies between the two models. We believe the main reason is that the RCC cell lines originate from different patients and belong to distinct RCC subtypes: ACHN (SETD2 WT) is classified as papillary renal cell carcinoma, whereas A498 (SETD2 mutant) is classified as clear cell renal cell carcinoma. SETD2 is not the only mutated gene between these cell lines, and their overall transcriptional profiles vary. In this context, the SETD2 mutation may be one of several factors contributing to enhanced cryptic initiation.

Although we also observe differences in alternative polyadenylation between the models (Fig 3F), the overall pattern of cryptic initiation induction in class I and class II genes, as well as the presence of readthrough transcription on class II genes, is consistent across both U2OS and RCC models. To highlight this point more clearly in the manuscript, we added the sentence: "However, since the RCC cell lines are derived from different patients, SETD2 activity might not be the only factor contributing to enhanced initiation." (page 8)

5. While Setd2 is well-documented for its role in Pol II elongation, the authors should propose a model explaining how it might also influence transcription initiation.

Additionally, experiments to demonstrate that initiation defects could be indirectly caused by known issues with Setd2 mutants, such as cryptic initiation or antisense transcription, would strengthen this hypothesis.

To determine which transcriptional effect occurs first – readthrough or the initiation defect – we now additionally performed POINT-seq on U2OS cells treated with the EPZ-719 inhibitor for 9 days. This time point was chosen because it was sufficient to detect readthrough by qPCR, while we did not observe any effects on U2OS cells with shorter treatment (3 days) with EPZ-719 despite the loss of H3K36me3 (Fig 4A-E). The new sequencing data confirmed the qPCR results: readthrough occurs after 9 days of SETD2 inhibition, but to a lower level compared to KO (Fig 4F).

At the same time, we did not observe a decrease in initiation levels in either class I or class II genes; in fact, we detected a slight increase in signal at the TSS for both classes (Fig 4F). This suggests that the termination defect occurs before any impairment in initiation.

Based on our mass spectrometry results from the nuclear fraction of U2OS cells (Fig 5A and EV5B), we hypothesize that the initiation defect may be an indirect consequence of a reduced pool of available RNAPII molecules. Because termination is inefficient, polymerases remain engaged on many genes for longer periods, limiting the number of free RNAPII molecules available for new rounds of transcription. We have incorporated this explanation into the manuscript (page 12): "Moreover, the inefficient initiation observed in SETD2 KO seems to be a long-term consequence that follows defective termination. Reduced initiation levels may result from a diminished pool of available RNAPII molecules (Fig 5A, EV5B) which remain engaged in the transcription of many genes for longer periods due to inefficient termination".

6. Investigating whether genes involved in osteosarcoma or ccRCC tumorigenesis are enriched in either Class I or Class II could provide insights into Setd2's disease-specific roles, enhancing our understanding of its function in cancer development.

We didn't find enrichment of genes involved in osteosarcoma or RCC tumorigenesis (see below).

From statistics point of view this isn't surprising as both gene classes are very large, it is not likely to find enrichments looking at >10% of genes.

Methods: genes involved in RCC/osteosarcoma progression were downloaded from DISGENET. Fisher's exact test was used (2x2 contingency table) to assess if there is any enrichment of cancer-related genes in both class I and class II genes.

Results:

For RCC

	In disease list (RCC)	Not in disease list
Class II genes	9	844
All genes	176	9286

p-value = 0.1046

odds ratio

0.5626488

	In disease list (RCC)	Not in disease list
Class I genes	58	4781
All genes	127	5349

p-value = 1.962e-05

odds ratio

0.5109832 (cancer-related genes are not enriched, rather depleted (< 1))

For osteosarcoma

	In disease list (osteo)	Not in disease list
Class II genes	1	1388
All genes	54	8808

p-value = 0.005096

odds ratio

0.1175267 (cancer-related genes are not enriched, rather depleted (<1))

	In disease list (osteo)	Not in disease list
Class I genes	21	4294
All genes	34	5902

p-value = 0.5867

odds ratio

0.8489672

Referee #3:

SETD2 deposits H3K36me3 in the bodies of active genes and is positively correlated with

transcription. Importantly, while the effects of SETD2 on transcription have been previously investigated, this did not include nascent transcription analysis as well as a direct assessment of engaged RNA PolII dynamics. Nakayama et al. address this gap in detail through a series of sophisticated techniques for measurement of RNA PolII-engaged RNA applied to mutant cancer cell lines.

The main conclusion is that SETD2 regulates both transcription initiation and termination, where it further can regulate cleavage as well as read-through. Most of the experimental evidence is based on two cancer types, namely i) a U2OS cell line in which SETD2 has been deleted and ii) two patient-derived RCC cell lines with and without a catalytic mutation of SETD2. While stable mutants and variable genetic backgrounds do not allow for conclusions regarding the direct effects of SETD2 loss, the authors acknowledge this limitation and additionally use a catalytic inhibitor to test their findings. This experiment corroborates the overall conclusions on the one hand, but also provides evidence for widespread indirect effects.

The addressed question is important and the authors' approach provides many interesting insights using state-of-the-art experimental methods, data and analysis with the highest quality. Similarly, the manuscript is very-well written, clear and concise, with clear figures that are appropriately interpreted.

While the manuscript's core strength lies in its thorough analysis and technical approach, as well as in the use of a combination of methods that actually allow to clarify the transcriptional effects, mechanistic conclusions cannot be fully drawn from the here chosen experimental system. My major concern is that the usage of stable mutant cell lines with variable parental background may interfere with causal conclusions and the direct effects of H3K36-methyltransferase inhibition remain unclear. I believe the points below will strengthen the main message of the manuscript.

Thank you for your thoughtful comments which helped us to improve our manuscript! We address your concerns below.

Major comments:

1. Usage of cancer cell lines as described here may be problematic for two reasons: a) RCC cell lines are derived from two different patients and therefore likely are both genetically and cell-type-wise not directly comparable, yet treated here as wildtype versus knockout. This makes the magnitude of the effects obtained in the RCC cell lines incomparable with the effects in the U2OS cell lines that have the same genetic background. Going forward, I recommend to the authors to downplay this comparison and acknowledge the limitation.

We agree with the referee and made this now clear in the text: "We need to stress here the limitation of our approach: the two RCC cell lines are derived from two different patients and therefore diverse both genetically and cell-type-wise. In contrast, U2OS WT and KO cells share a common background. As a result, the magnitude of the effects observed in the RCC and U2OS systems cannot be directly compared." (page 7 in current manuscript). We also downplayed and qualified the comparison between the two cell lines in further sections of the paper.

b) The knockout/mutation are stable, therefore the effects are likely to be indirect. To strengthen the main message, it would be important to investigate direct effects beyond the effects reported in Fig. 4 (see below).

To investigate direct effects, we performed POINT-seq experiments on U2OS cells treated with EPZ-719 inhibitor on day 9 (described in answer to Referee #1, point 5). We also checked RDT index for a few representative genes in non-cancerous cell lines (hTERT-RPE and HEK293 cells) after EPZ-719 treatment (described in the comment of Referee #2, point 3). Both sets of experiments, shown in

revised Fig 4 and EV4, confirm that treatment of cells with SETD2 inhibitor leads to readthrough transcription on class II genes. On the other hand, the drop in transcription initiation appears to be a downstream effect. We now discuss this on page 12 of our revised manuscript: “the inefficient initiation observed in SETD2 KO seems to be a long-term consequence that follows defective termination. Reduced initiation levels may result from a diminished pool of available RNAPII molecules (Fig 5A, EV5B) which remain engaged in the transcription of many genes for longer periods due to inefficient termination”.

2. Direct effects are tested by using a methyltransferase inhibitor in Fig. 4, whereby unsurprisingly the effects achieved at tested genes are smaller than in the stable cell lines. Through investigating the proteome, the authors then conclude that the majority of SETD2-mediated effects on transcription are indirect, since the inhibitor affects RNA PolII and other RNA processing enzymes at the protein level.

We respectfully point out your misunderstanding – the proteome was analyzed in SETD2 KO versus wild-type U2OS cells.

To contextualize these effects and discuss the potential indirect effects further, the authors should address the following two points:

a) How does SETD2 inhibition affect the proteins responsible for RNA processing? Is there a direct effect upon methyltransferase activity inhibition on the readthrough/termination of the genes encoding proteins involved in RNA processing/RNA PolII?

It appears that having put qPCR data from inhibitor treatment, and mass-spec analysis of SETD2 KO cells in one single figure has led to confusion. Therefore, in the revised manuscript we split them in separate Fig 4 and 5.

The referee asked for evidence whether the SETD2-depletion effect we see on protein level (i.e. in the knock-out cells) might be due to readthrough on the genes encoding the proteins involved in RNA processing.

To answer this, we checked for changes in nascent transcription levels in the gene body, and in the readthrough region (downstream of the annotated PAS) in SETD2 KO versus wild-type U2OS cells. We selected the 10 most downregulated proteins in each category – initiation, elongation, termination, and splicing (Fig EV5B). In SETD2 KO cells, we found that roughly half of the genes encoding these proteins are also downregulated at the transcriptional level. For the remaining genes, we observed an increase in signal in the +10 kb region relative to the gene body (e.g., SNRPE, CSTF1, BRD4, FIP1L1, CTR9), which indicates readthrough transcription (Fig EV5C).

We performed the same analysis in cells treated with EPZ-719 on day 9. As expected, transcriptional changes were much less pronounced than in U2OS SETD2 KO cells. Nevertheless, a clear cluster of genes at the bottom of the heatmap shows an emerging readthrough (Fig EV5D and E).

We thank the referee for this mechanistically insightful suggestion. The extended analysis is now demonstrated in the revised manuscript in EV5.

b) Measure H3K36me3 before/after treatment to test if the observed effects on cleavage/readthrough correlate with alterations in the methyltransferase activity and are therefore likely to be direct.

The observed effects on cleavage/readthrough are likely indirect.

In U2OS, although H3K36me3 becomes almost undetectable after 3 days of treatment (Fig 4A), our qPCR analysis shows no readthrough induction at this time point (Fig 4C–E). In this model, readthrough appears only after 9 days of treatment (Fig 4C–E).

We now also tested this in non-cancerous cells – HEK293 and hTERT-RPE – and included additional time points for the latter. Interestingly, in hTERT-RPE cells the effect appears earlier than in U2OS, with

clear readthrough induction after 3days (72 hours) of treatment (Fig EV4B–D). As H3K36me3 levels drop significantly after 24 hours of treatment in this cell line, this delay in readthrough induction again suggests an indirect mechanism. However, for CD58 and COP1, we observe a slight increase in the RDT index after 24 hours and 48 hours of treatment, respectively.

Together, these results highlight that the impact of SETD2 methyltransferase activity on transcription varies across cell types and even among individual genes.

Minor comments:

3. Were the genome-wide experiments calibrated (using spike-ins)? If not, this could distort potential global effects.

Our U2OS POINT-seq libraries contain spike-ins (S2R cells derived from *Drosophila*) and we were planning to normalize read counts based on those. However, unfortunately due to technical reasons replicate 2 and 3 were not properly spiked-in (table below).

U2OS	S2R	Human	% of S2R reads
WT1	5154653	85893695	5.661
WT2	32256	75772371	0.043
WT3	22827	81736136	0.028
KO1	3242126	54341301	5.630
KO2	41036	69043054	0.059
KO3	36366	70110570	0.052

Replicate 1, where the spike-in 5% rate was successful, was normalized based on spike-in (graphs pasted below). Since the metaplot which takes into account spike-ins is very similar and gives the same conclusions at the one normalized based on RPM (Fig 1C-D), so we decided to normalize all POINT-seq replicates to RPM.

Class I:

Class II:

We did not employ spike-ins for 3'mRNA-seq, however performed very thorough statistical analysis and in the revised version also added qPCR as an additional verification strategy (Fig EV3G-I).

4. Typo on p.9, paragraph heading: 'The effects of SETD2 activity on transcription readthrough is...' should be 'are'

Thank you, we have now corrected the typo.

Dear Dr. Kamieniarz-Gdula,

Thank you for your patience while your revised manuscript was re-reviewed. We have now received the enclosed reports from the referees, as well as cross-comments. As you will see, while referee 2 is more critical and raises important points, both referees 1 and 3 agree that these can be addressed in the ms text and that your study can be accepted for publication. Please address all remaining comments in the ms text and please also submit a point-by-point response to all final comments. I agree with referee 3 that "aids gene definition" should probably be deleted from the ms title.

A few editorial requests will also need to be addressed before we can proceed with the official acceptance of your manuscript:

- Please place the Disclosure and Competing Interest Statement to after the Acknowledgements.
- There is one author name discrepancy: Michal Gdula in our online system vs Michał R. Gdula in the manuscript, please correct. The corr. authors' emails need to be added to the cover page.
- The authors credits need to be removed from the ms file. All credits need to be entered during online ms submission.
- In the author checklist, all questions on the statistics need to be answered, please send us a new, completed author checklist.
- The funding info in our online system and the ms file need to be kongruent. This Funder name is different: [2018/30/E/NZ1/00073], please correct. The funders listed in the Comments box online need to be removed and inserted as separate funder entries; it is fine if the system does not recognize a funder, the information just needs to match.
- Table EV1 is a dataset and needs to be updated to Dataset EV1 in all places (file name, callouts, etc); Table EV2 is an EV table and should then be updated to Table EV1 in all places. Table EV1 could also be part of the Reagents and Tools table, which might make it easier for the reader.
- Please add specific URLs to the Data Availability Section that link directly to the actual deposited data that need to be freely accessible upon the online publication of your ms.
- The Expanded View figure legends need the subheading "EV figure legends" and the EV Figures need to be called Figure EV1, EV2 etc.
- Materials and Methods should be just Methods.

* Figure Legends - Comments *

- Please define the annotated p values ****/**/*/* as well as provide the exact p-values for the same in the legend of figure EV 1k, l, n, p, t, w as appropriate and reasonable.
- Please note that the exact p values are not provided in the legends of figures 1g, m; 3d, e. Please provide exact values as reasonable.
- Please indicate the statistical test used for data analysis in the legends of figures 5a; EV 1k, l, n, p, t, w; EV 5a
- Please note that the box plots need to be defined in terms of minima, maxima, centre, bounds of box and whiskers, and percentile in the legends of figures 2e, f; EV 1k, l, n, p, t, w; EV 2b
- Please note that the box plots need to be defined in terms of minima, maxima, bounds of box and whiskers, and percentile in the legends of figures 1g, m; 3d, e
- Please note that information related to n is missing in the legends of figures EV 1k, n, t, w"

* Data Citation - Comments *

- Please note that the data citations are not tagged with the label "DATASET" in the reference list, please correct.

EMBO press papers are accompanied online by A) a short (1-2 sentences) summary of the findings and their significance, B) 2-3 bullet points highlighting key results and C) a synopsis image that is exactly 550 pixels wide and 200-600 pixels high (the height is variable). The synopsis image should provide a sketch of the major findings, like a graphical abstract. Please note that text needs to be readable at the final size. Please send us this information along with the final manuscript.

Referee #1:

The authors carefully addressed reviewers comments, and I believe that authors have generated nice data that need to be published. Authors have performed extensive additional analyses that strengthen conclusions of the manuscript and eliminated some inconsistencies present in the original version. Including analyses of the effect of EPZ-719 treatment on hTERT-RPE cell line was very informative as well as analyses delineating characteristic features of the Class I and II genes affected by Setd2.

Including analyses of the cleavage efficiency on the genes with single PAS was also beneficial. However, I wonder whether it is a bit premature to state that APA is unrelated to RT without assessing whether efficiency of the cleavage on proximal PAS is comparable to efficiency of cleavage at the distal PAS that is used when setd2 is present? Perhaps this could be re-phrased? Nevertheless, overall, the findings in this manuscript are very valuable and should be published, but I believe the model should be adjusted.

Referee #2:

The revised manuscript still does not adequately address the main concerns raised in the initial review. Although the study employs state-of-the-art sequencing approaches to profile transcriptomic changes in SETD2-dysfunctional tumor cells, the results remain largely descriptive and lack convincing mechanistic insight. Moreover, the continued reliance on multiple tumor cell lines is concerning, as their diverse genetic backgrounds introduce substantial confounding variables that weaken the reliability and interpretability of the genomic analyses. This reviewer also remains unconvinced by the attempt to assess the "direct effects" of SETD2 catalytic loss using POINT-seq, as the 9-day post-EPZ-719 treatment time point is intrinsically problematic for disentangling primary versus secondary effects. While the reviewer appreciates the authors' transparency in acknowledging this limitation, the additional experiments themselves demonstrate that many of the purportedly "surprising" or "novel" findings are in fact driven by indirect effects, which significantly diminishes the overall impact of the study.

Collectively, these results further reinforce the concern that the manuscript primarily presents phenomenology, with limited predictive or mechanistic power regarding the indirect consequences of SETD2 dysfunction. As highlighted in the authors' response, "We interpret the limited overlap between the models to be a result of osteosarcoma and RCC cells having different transcriptional profiles, and class II genes being associated with higher expression levels." Furthermore, the authors explicitly acknowledge that "However, since the RCC cell lines are derived from different patients, SETD2 activity might not be the only factor contributing to enhanced initiation." Together, these statements underscore the extent to which confounding variables, rather than SETD2-specific mechanisms, likely drive many of the observed effects.

Referee #3:

I thank the authors for thorough addressing of my concerns and congratulate them on this interesting study.

Cross-comments from referee 3:

Reviewer 2 raises two important concerns: usage of multiple cancer cell lines and the indirect nature of the observed effects.

My impression is that the cell lines' variability has been adequately addressed and I agree with the authors that different effects may be expected in cells with different transcriptional/epigenetic profiles.

I agree with the reviewer that the effects are indirect and it would be undoubtedly interesting to determine what mediates the readthrough effect (maybe the components of the transcriptional machinery that are downregulated at the proteomic level or another protein specific enough for such an effect?). To their credit, this point has been acknowledged by the authors and I still think that the study is interesting enough to be published. Potentially the title could be adjusted.

Cross-comments from referee 1:

I also agree on those two points. Delineating direct effects of course would be great but I think that this goes beyond the scope of this work. Overall, the authors have done extensive efforts to address the reviewers comments. This study reports effects on transcription caused by the depletion of SETD2 that have not been previously reported and its is important for the field to know and therefore I believe the revised manuscript should be published.

Referee #1:

The authors carefully addressed reviewers comments, and I believe that authors have generated nice data that need to be published. Authors have performed extensive additional analyses that strengthen conclusions of the manuscript and eliminated some inconsistencies present in the original version. Including analyses of the effect of EPZ-719 treatment on hTERT-RPE cell line was very informative as well as analyses delineating characteristic features of the Class I and II genes affected by Setd2.

Including analyses of the cleavage efficiency on the genes with single PAS was also beneficial. However, I wonder whether it is a bit premature to state that APA is unrelated to RT without assessing whether efficiency of the cleavage on proximal PAS is comparable to efficiency of cleavage at the distal PAS that is used when setd2 is present? Perhaps this could be re-phrased? Nevertheless, overall, the findings in this manuscript are very valuable and should be published, but I believe the model should be adjusted.

We thank the reviewer again for his/her kind words and the constructive criticism which helped us to improve the manuscript.

Regarding the suggestion of rephrasing, the following results reported in the manuscript suggest that readthrough is not connected to APA:

- 1) widespread readthrough occurs in both U2OS and ccRCC; in contrast APA is widespread only for U2OS (majority of genes), while in ccRCC the overwhelming majority of genes does not undergo APA (Figure 3F);*
- 2) in U2OS, where readthrough and APA co-occur, the major direction of APA is proximal, hence opposite to the direction of readthrough (Figure 3F, EV3G);*

Those two very clear tendencies in our data strongly suggest a disconnection between readthrough and APA. We agree this was unexpected, also to us. However, given those unexpected and unexplained, yet clear and reproducible effects of SETD2 depletion - we would prefer to keep the current wording in the manuscript body. We hope in this way we can spur further research into the mechanisms behind this paradox, especially by laboratories using different experimental approaches. Nevertheless, we rephrased the wording of the abstract to the following, weaker statement: "Additionally, alternative polyadenylation upon SETD2 activity loss is highly cell type specific, and no relationship with transcription readthrough was observed."

We also adjusted the title to: "SETD2 methyltransferase activity promotes correct transcription initiation and termination"

Referee #2:

The revised manuscript still does not adequately address the main concerns raised in the initial review. Although the study employs state-of-the-art sequencing approaches to profile transcriptomic changes in SETD2-dysfunctional tumor cells, the results remain largely descriptive and lack convincing mechanistic insight. Moreover, the continued reliance on multiple tumor cell lines is concerning, as their diverse genetic backgrounds introduce substantial confounding variables that weaken the reliability and interpretability of the

genomic analyses. This reviewer also remains unconvinced by the attempt to assess the "direct effects" of SETD2 catalytic loss using POINT-seq, as the 9-day post-EPZ-719 treatment time point is intrinsically problematic for disentangling primary versus secondary effects. While the reviewer appreciates the authors' transparency in acknowledging this limitation, the additional experiments themselves demonstrate that many of the purportedly "surprising" or "novel" findings are in fact driven by indirect effects, which significantly diminishes the overall impact of the study.

Collectively, these results further reinforce the concern that the manuscript primarily presents phenomenology, with limited predictive or mechanistic power regarding the indirect consequences of SETD2 dysfunction. As highlighted in the authors' response, "We interpret the limited overlap between the models to be a result of osteosarcoma and RCC cells having different transcriptional profiles, and class II genes being associated with higher expression levels." Furthermore, the authors explicitly acknowledge that "However, since the RCC cell lines are derived from different patients, SETD2 activity might not be the only factor contributing to enhanced initiation." Together, these statements underscore the extent to which confounding variables, rather than SETD2-specific mechanisms, likely drive many of the observed effects.

We thank the reviewer for their suggestions that allowed us to improve our manuscript. We acknowledge the limitations of our work.

At the same time, we note that the aims and scope of EMBO Reports according to the Journal's website are the following: "EMBO Reports publishes both long- and short-format papers in all areas of molecular-, cell- developmental biology and ecology. The journal welcomes studies that either confirm or refute prominent claims in the literature, as well as null data on important, open questions in the biosciences. [...] Editors select manuscripts on the basis of technical quality and a combination of conceptual advance, general interest and physiological relevance, rather than the level of mechanistic detail reported." In our view, and view of referees #1 and #3 our work fulfills the aims and scope of the Journal.

In order to be entirely upfront with the readers that the effects we observe in our study are not primary, but indirect, we have changed the last sentence of our abstract to the following: "We demonstrate that methyltransferase activity of SETD2 stimulates proper initiation, prevents cryptic initiation and promotes efficient 3' end processing, however it does so indirectly."

Referee #3:

I thank the authors for thorough addressing of my concerns and congratulate them on this interesting study.

Thank you!

Cross-comments from referee 3:

Reviewer 2 raises two important concerns: usage of multiple cancer cell lines and the indirect nature of the observed effects.

My impression is that the cell lines' variability has been adequately addressed and I agree with the authors that different effects may be expected in cells with different transcriptional/epigenetic profiles.

I agree with the reviewer that the effects are indirect and it would be undoubtedly interesting to determine what mediates the readthrough effect (maybe the components of the transcriptional machinery that are downregulated at the proteomic level or another protein specific enough for such an effect?). To their credit, this point has been acknowledged by the authors and I still think that the study is interesting enough to be published. Potentially the title could be adjusted.

Following on those comments, to underscore the indirect nature of our findings, we now made this clear in the last sentence of the abstract.

Additionally, we adjusted the title to: "SETD2 methyltransferase activity promotes correct transcription initiation and termination"

Cross-comments from referee 1:

I also agree on those two points. Delineating direct effects of course would be great but I think that this goes beyond the scope of this work. Overall, the authors have done extensive efforts to address the reviewers comments. This study reports effects on transcription caused by the depletion of SETD2 that have not been previously reported and its is important for the field to know and therefore I believe the revised manuscript should be published.

We agree. Thank you!

Dr. Kinga Kamieniarz-Gdula
Adam Mickiewicz University
Center for Advanced Technologies
Poland

Dear Kinga,

I am very pleased to accept your manuscript for publication in the next available issue of EMBO reports. Thank you for your contribution to our journal.

You may qualify for financial assistance for your publication charges - either via a Springer Nature fully open access agreement or an EMBO initiative. Check your eligibility: <https://link.springer.com/journal/44319/how-to-publish-with-us>

>>> Please note that it is EMBO Reports policy for the transcript of the editorial process (containing referee reports and your response letter) to be published as an online supplement to each paper. If you do NOT want this, you will need to inform the Editorial Office via email immediately. More information is available here: <https://link.springer.com/partners/embo-press/editorial-policies#Peer%20review>